# THE ACHILLES' HEEL OF LLMs: HOW ALTERING A HANDFUL OF NEURONS CAN CRIPPLE LANGUAGE ABILITIES

**Zixuan Qin[1], Qingchen Yu[2,3], Kunlin Lyu[1], Zhaoxin Fan[2,3*], Yifan Sun[1*]**
[1]Center for Applied Statistics, School of Statistics, Renmin University of China
[2]Beijing Advanced Innovation Center for Future Blockchain and Privacy Computing
[3]School of Artificial Intelligence, Beihang University

## ABSTRACT

Large Language Models (LLMs) have become foundational tools in natural language processing, powering a wide range of applications and research. Many studies have shown that LLMs share significant similarities with the human brain. Neuroscience research has found that a small subset of biological neurons in the human brain are crucial for core cognitive functions, which raises a fundamental question: do LLMs also contain a small subset of critical neurons? In this paper, we investigate this question by proposing a Perturbation-based Causal Identification of Critical Neurons method to systematically locate such critical neurons in LLMs. Our findings reveal three key insights: (1) LLMs contain ultra-sparse critical neuron sets. Disrupting these critical neurons can cause a 72B-parameter model with over 1.1 billion neurons to completely collapse, with perplexity increasing by up to 20 orders of magnitude; (2) These critical neurons are not uniformly distributed, but tend to concentrate in the outer layers, particularly within the MLP down_proj components; (3) Performance degradation exhibits sharp phase transitions, rather than a gradual decline, when these critical neurons are disrupted. Through comprehensive experiments across diverse model architectures and scales, we provide deeper analysis of these phenomena and their implications for LLM robustness and interpretability. These findings can offer guidance for developing more robust model architectures and improving deployment security in safety-critical applications. Our code is available at `https://github.com/qqqqqqqzx/The-Achilles-Heel-of-LLMs`.

## 1 INTRODUCTION

Understanding the fundamental principles underlying intelligent computation has long attracted researchers across neuroscience and artificial intelligence. A growing body of evidence suggests that the brain's computational power may critically depend on a small fraction of highly influential neurons, rather than being uniformly distributed across all neural elements. This principle, stated by Arshavsky (2001), suggests that sparsely distributed, genetically distinct neurons function as computational bottlenecks, each having a major influence over specific brain functions.

In parallel, Large Language Models (LLMs)—with their billions of parameters—have demonstrated capabilities mirroring human cognition, from language comprehension to problem-solving (Riztha et al., 2024; Niu et al., 2024). Recent research shows clear parallels: the hierarchical processing and predictive coding mechanisms in LLMs align closely with the brain's layered neural architectures (Schrimpf et al., 2021; Caucheteux & King, 2022; Mischler et al., 2024), hinting that these artificial systems may emulate biological computation. This observation leads to a key question:

**Q: Do LLMs likewise possess a small set of critical artificial neurons that are indispensable for their capabilities?**

---

*Project leaders and Corresponding authors. Emails: zhaoxinf@buaa.edu.cn, sunyifan@ruc.edu.cn

In the field of artificial intelligence, previous work has used perturbation-based methods to identify important model components (Hanna et al., 2023; Zhang & Nanda, 2024). To explore this key question, we apply perturbation-based analysis to identify critical neurons. Our goal is to find a small subset of neurons whose collective disruption causes catastrophic system-wide collapse.

To put this investigation into practice, we introduce the Perturbation-based Causal Identification of Critical Neurons method, as illustrated in Figure 1. Given any text as input, the proposed method consists of a two stage optimization process to locate the so called critical neurons. In the first stage, the method injects controlled noise into the model's input and measures the resulting activation differences across neurons, thereby generating a ranked list of candidates most likely to influence model behavior. In the second stage, we sequentially mask these top-ranked neurons in a greedy manner, closely monitoring changes in model perplexity. This allows us to causally determine which neurons are truly essential for the model's function.

Using this method, we analyze a diverse set of datasets—including Neurons, WikiText-103, C4, MMLU-Pro (Wang et al., 2025), IFEval (Zhou et al., 2023), GPQA-Diamond (Rein et al., 2024), HumanEval (Peng et al., 2024), MATH (Hendrycks et al., 2021), MGSM (Shi et al., 2023), and SimpleQA (Wei et al., 2024)—as well as multiple LLMs such as Llama-3 (Grattafiori et al., 2024), Gemma (Gemma Team, 2024), DeepSeek-R1 (DeepSeek-AI, 2025), Phi-3, Phi-3.5 (Abdin et al., 2024), and Qwen2.5 (Qwen, 2025). According to our experiment, we observe consistent and interesting patterns across all settings. In particular, our findings reveal several key phenomena. First, we discover that LLMs are governed by an ultra-sparse set of critical neurons: astonishingly, disabling as few as three neurons can catastrophically impair a 72B-parameter model comprising over 1.1 billion neurons, driving its perplexity up by as much as 20 orders of magnitude. Second, we find that these critical neurons are not randomly distributed throughout the network, but are instead highly concentrated in the outer layers, particularly within the MLP down_proj components, which is consistent with the observations reported in Yu et al. (2025). Third, we observe that performance degradation does not occur gradually, but rather through sharp phase transitions. In section 4, we provide a detailed discussion of these findings and hope these discoveries can help the community better optimize LLM architecture, training, and inference processes. Our contributions are summarized as follows:

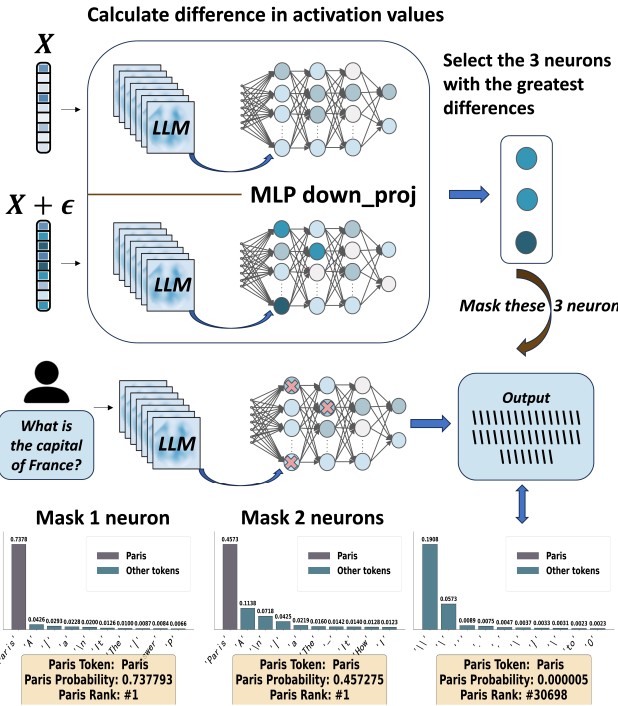

Figure 1: Illustration of critical neuron identification and progressive masking effects, using DeepSeek-R1-Distill-Llama-70B as an example. The top panel shows that our method identifies 3 critical neurons located in MLP down_proj components of the transformer architecture. The bottom panel demonstrates the progressive degradation of model performance on the question "What is the capital of France?" through sequential masking: the left chart shows token probabilities after masking the first critical neuron (Paris probability: 0.737793, rank 1), the middle chart shows the effect of masking the first two critical neurons (Paris probability: 0.457275, rank 1), and the right chart reveals catastrophic failure after masking all three critical neurons (Paris probability: 0.000005, rank 30698). The progression illustrates a sharp phase transition where masking the complete critical neuron set triggers sudden collapse rather than gradual degradation.

- We propose a systematic method that integrates noise-based sensitivity analysis with causal verification, which enables us precisely to identify critical neurons that are indispensable for the overall functionality of LLMs. Our method shows exceptional robustness, consistently identifying identical critical neurons for any input texts exceeding a minimum token length threshold ($T > 10$).

- Our analysis uncovers several surprising findings: (a) model performance is predominantly governed by a remarkably small subset of critical neurons; (b) these crucial neurons tend to cluster within specific outer-layer MLP components; and (c) performance degradation exhibits sharp phase transitions if we drop these neurons, rather than a gradual decline, when these critical neurons are disrupted.

- We conduct comprehensive experiments across 21 models ranging from 0.5B to 72B parameters, spanning multiple architecture families and evaluated on diverse datasets including WikiText-103, C4, and seven downstream benchmarks. Across all these varied experimental conditions, our results demonstrate remarkable consistency, with identical critical neuron patterns emerging regardless of model scale, architecture design, or evaluation dataset. This high degree of consistency across such extensive and diverse experimental settings shows the fundamental and universal nature of critical neurons in LLMs.

## 2 RELATED WORK

**Critical Components Analysis in LLMs.** Recent research has revealed a variety of vulnerabilities in LLMs, but most studies have focused on parameter-level or component-level rather than neuron-level analysis. For example, Kovaleva et al. (2021) investigates weight outliers in LLMs and finds that anomalous parameters arising during pre-training can significantly affect performance when disabled. Bondarenko et al. (2021) explores activation outliers that influence attention mechanisms, particularly in processing special tokens. Sun et al. (2024) uncovers persistent, large-scale activation patterns at fixed positions across transformer layers, and Yang et al. (2024) attributes such phenomena to architectural components like gated linear units. At the same time, studies on "super weights" show that individual parameters in MLP down_proj layers can catastrophically impact model performance (Yu et al., 2025). Our work goes beyond these studies by introducing a neuron-level perspective, identified the "critical neurons", and ruled out the possibility that they were scale outliers. We propose a Perturbation-based Causal Identification framework to systematically locate these critical neurons. Though our work may appear similar to that of Yu et al. (2025), it is fundamentally different in the following aspects: (1) Disruption strength: Their weight-level perturbations only reduce accuracy, whereas our neuron-level lesions cause a complete loss of language capability; (2) Methodological applicability: Their identification process requires manual observation, while our automated Perturbation-based Causal Identification framework provides a reproducible and engineering-ready pipeline; (3) Multi-neuron interaction: Their analysis does not examine interaction effects, whereas we reveal a phase-transition phenomenon triggered only by collective neuron disruption; (4) Research unit: They study parameters, whereas we analyze neurons; (5) Discovery process: Our findings emerge from analyzing functional collapse patterns, which reveal sharp phase transitions when coordinated neuron groups are disrupted.

**Neurons Localization and Editing.** Recent studies have advanced neuron-level localization and editing in LLMs, showing that neural parameters can be divided between pattern learning and memorization (Bender et al., 2021). Targeted interventions on specific neurons or clusters enable systematic modification of factual knowledge (Meng et al., 2022; Dai et al., 2022; Li et al., 2024b), control of personal information memorization (Chen et al., 2024), multilingual capabilities (Tang et al., 2024), and even safety constraints (Zhou et al., 2025). Recent work on "super-neurons" (Gong et al., 2025) instead focuses on neurons with extreme polysemantic load, showing that amplifying their activations can strongly steer behavior, while masking them has only a modest effect. Related recent work on "Wasserstein neurons" (Kong et al., 2025) identifies neurons that encode syntactic structure via non-Gaussian negative pre-activations and demonstrates that disrupting many such neurons selectively impairs syntactic competence. Further, behavioral attributes such as "aggression" can be manipulated by localizing and editing a small set of neurons (Lee et al., 2025), and polysemantic neurons in fusion heads are linked to modality collapse in multimodal models (Chaudhuri et al., 2025). However, existing work primarily focuses on task-specific interventions, lacking systematic frameworks for identifying neurons critical to core model functionality. Our work addresses this gap by introducing a principled method for neuron localization and causal verification.

## 3 METHODOLOGY

This paper aims to define and address the problem of identifying **critical neurons** in LLMs. The central question is: *Which neurons are essential for the model's performance, such that their removal causes substantial degradation?* Identifying such neurons enables us to better understand model internals and analyze the consequences of their ablation.

Formally, let $N$ denote the set of all neurons in the model. Our goal is to find a sparse subset $S^* \subseteq N$ (with $|S^*| \ll |N|$) such that masking these neurons results in a significant drop in model performance, as measured by a predefined degradation threshold $\epsilon$.

To operationalize this, we apply a neuron masking protocol: for any neuron $(l, i)$ at layer $l$ and index $i$, we define its activation under masking as

$$\tilde{n}_l^{(i)}(\mathbf{x}) = \begin{cases} 0 & \text{if } (l, i) \in S \\ n_l^{(i)}(\mathbf{x}) & \text{otherwise} \end{cases} \tag{1}$$

where $S$ is the set of masked neurons. Let $M^{-S}$ denote the model with neurons in $S$ masked.

We quantify the impact of masking via the change in perplexity (Appendix A.1) given a sequence of data an input:

$$\Delta(\mathbf{x}, S) = \log_{10}\left(\frac{\text{PPL}_{M^{-S}}(\mathbf{x})}{\text{PPL}_M(\mathbf{x})}\right) = \frac{1}{T \ln 10} \sum_{t=1}^{T} [\log P_M(x_t|\mathbf{x}_{<t}) - \log P_{M^{-S}}(x_t|\mathbf{x}_{<t})] \tag{2}$$

where $T$ is the sequence length, PPL denotes perplexity, and $P_M$ is the predicted token probability.

The critical neuron identification problem is thus:

*Given a model $M$ and input $\mathbf{x}$, find the minimal neuron set $S^* \subseteq N$ such that $\Delta(\mathbf{x}, S^*) \geq \epsilon$.*

After identifying $S^*$, we can further analyze the effects of masking these neurons to understand their functional roles and contributions to model behavior. To solve the problem, building on the mathematical framework established in Appendix A.1, we propose a method, which operationalizes the theoretical definitions through a two-stage process: (1) neuron importance quantification via sensitivity analysis, and (2) causal verification of criticality through systematic masking interventions. Next, we introduce each stage in detail. Additional computational efficiency analysis and algorithmic implementations are provided in Appendix A.2.

**Stage 1: Neuron Importance Evaluation.** The first stage establishes neuron ranking based on sensitivity to input perturbations. Given input text $p$ and model $f_\theta$, we embed $p$ into sequence $\mathbf{x}$, and employ Monte Carlo sampling with $K$ iterations to estimate each neuron's importance. In each iteration $i$, we generate a perturbed input $\tilde{\mathbf{x}}_i = \mathbf{x} + \alpha \cdot \boldsymbol{\epsilon}_i$ where $\boldsymbol{\epsilon}_i \sim \mathcal{N}(\mathbf{0}, \mathbf{I})$ represents Gaussian noise and $\alpha$ controls the perturbation magnitude. We then compute the noisy activation $A_i^{\text{noisy}} = f_\theta(\tilde{\mathbf{x}}_i)$ and compare it with the clean activation $A^{\text{clean}} = f_\theta(\mathbf{x})$.

For each neuron $s$, we iteratively accumulate the activation differences across all $K$ perturbations to compute the importance score:

$$\text{Imp}(s) = \frac{1}{K} \sum_{i=1}^{K} |A_s^{\text{clean}} - A_{i,s}^{\text{noisy}}| \xrightarrow{K \to \infty} \mathbb{E}_{\boldsymbol{\epsilon} \sim \mathcal{N}(\mathbf{0}, \mathbf{I})}[|f_{\theta,s}(\mathbf{x}) - f_{\theta,s}(\mathbf{x} + \alpha \cdot \boldsymbol{\epsilon})|] \tag{3}$$

By the Law of Large Numbers, this Monte Carlo estimator converges to the true expected sensitivity as $K$ increases. After computing importance scores for all neurons, we sort them in descending order to obtain the ranked list $S = \{s_1, s_2, \ldots, s_N\}$ where $\text{Imp}(s_1) \geq \text{Imp}(s_2) \geq \ldots \geq \text{Imp}(s_N)$.

**Stage 2: Critical Neuron Identification.** The second stage solves the optimization problem of finding the minimal critical set. Given the importance-ranked neuron list $S$ from Stage 1 and threshold $\epsilon$, our objective is to find:

$$n^* = \arg\min_n \{n : \Delta(\mathbf{x}, S_n) \geq \epsilon\} \quad \text{where } S_n = \{s_1, s_2, \ldots, s_n\} \tag{4}$$

Since exhaustive enumeration of all possible subsets requires $O(2^{|N|})$ complexity, we leverage our importance ranking to design a greedy search algorithm. Given the sorted neuron list $S$ and threshold $\epsilon$ as inputs, we use an iterative greedy search that progressively increases the candidate set size as the iteration variable. Specifically, we iterate over $n = \Delta n, 2\Delta n, 3\Delta n, \ldots$ where $\Delta n$ is the step size, and in each iteration we update the candidate set $S_n = \{s_1, s_2, \ldots, s_n\}$ to include the top-$n$ highest-ranked neurons from our importance list. For each updated candidate set, we apply the masking operation by setting $\tilde{n}_l^{(i)}(\mathbf{x}) = 0$ for all neurons $(l, i) \in S_n$, then evaluate the resulting masked model $M^{-S_n}$ to compute the performance degradation $\Delta(\mathbf{x}, S_n) = \log_{10}(\text{PPL}_{M^{-S_n}}(\mathbf{x})/\text{PPL}_M(\mathbf{x}))$. The algorithm terminates when the degradation exceeds our threshold $\Delta(\mathbf{x}, S_n) \geq \epsilon$, returning the minimal set $S_{n^*}$ that causes catastrophic language capability degradation. This greedy approach reduces computational complexity to $O(|N|)$ by testing only $\lceil |N|/\Delta n \rceil$ candidate sets rather than all $2^{|N|}$ possible combinations, while effectively approximating the optimal solution.

Our method only takes one sequence as input for any LLMs, which makes our method extremely robust and reliable (see Section 4 experiment for proof). Under sufficient parameter settings (when $K$ and $\alpha$ are adequately large) and for input texts exceeding a minimum token length threshold ($T > 10$), the identified critical neurons remain identical across repeated experiments (details can be found in Section 4.4).

## 4 EXPERIMENTS

### 4.1 EXPERIMENTAL SETUP

**Datasets.**  We conduct our investigation across multiple datasets to assess both language modeling capabilities and downstream task performance. For language modeling, we primarily use WikiText-103 and the C4 dataset. To evaluate the impact of critical neuron masking on specific capabilities, we employ seven additional benchmarks: MMLU Pro (Wang et al., 2025) for advanced multi-domain knowledge, IFEval (Zhou et al., 2023) for instruction following, GPQA Diamond (Rein et al., 2024) for graduate-level scientific reasoning, HumanEval (Peng et al., 2024) for code generation, MATH (Hendrycks et al., 2021) for mathematical problem solving, SimpleQA (Wei et al., 2024) for factual question answering, and MGSM (Shi et al., 2023) for multilingual mathematical reasoning. Detailed evaluation protocols and metrics for each benchmark are provided in Appendix A.3.

**Implementation Details.**  We follow the methodology described in Section 3, with parameter configurations chosen to balance accuracy and computational efficiency. For input perturbation, we set the noise scale to $\alpha = 5$ and randomly generate $K = 100$ noisy samples for neuron importance evaluation.A greedy search with step size $\Delta n = 1$ and criticality threshold $\epsilon = 1$ is used throughout our analysis (see Appendix A.4). For each experiment, we use a consistent input text $p$: "Later reviews were more positive..." - a 30-token Wikipedia text that ensures standardized evaluation conditions across all models. All experiments are conducted on a high-performance computing cluster equipped with 4 NVIDIA A800 GPUs (80GB each). We use PyTorch (Paszke et al., 2019) and employ mixed-precision (fp16) for efficient memory utilization and faster inference. Model loading and inference are fully automated using standardized scripts. Our experimental evaluation is structured around three core components: (1) A comprehensive analysis revealing the existence, distribution, and failure patterns of ultra-sparse critical neurons across all models; (2) Comparison studies benchmarking our method against alternative neuron location strategies; (3) Ablation study assessing the robustness of our critical neuron localization algorithm. Extensive results validate our key findings regarding critical neuron vulnerabilities in modern LLMs.

### 4.2 INVESTIGATION RESULTS

Our investigation covers 21 LLMs spanning diverse architectures and parameter scales (0.5B–72B), including the Llama-3, Gemma, DeepSeek-R1-Distill, Phi-3, and Qwen2.5 families (Grattafiori et al., 2024; Qwen, 2025; DeepSeek-AI, 2025; Gemma Team, 2024; Abdin et al., 2024), and we conduct evaluations using multiple datasets. This investigation enables a systematic assessment of neuron criticality across a broad range of model architectures and parameter scales. Our investigation leads to the following three key findings. Next, we analyze the three findings in detail.

> **Key Findings**
>
> **Finding 1:** Disrupting as few as three ultra-sparse critical neurons can catastrophically impair LLM capabilities; for example, masking just 3 out of over 1.1 billion neurons in a 72B-parameter model leads to a complete collapse, with perplexity increasing by up to 20 orders of magnitude.
>
> **Finding 2:** Critical neurons are unevenly distributed across the network, exhibiting a strong tendency to cluster in the outer layers, particularly within MLP `down_proj` components.
>
> **Finding 3:** Performance degradation manifests as sharp phase transitions rather than gradual declines when these critical neurons are disrupted.

Table 1: Critical neuron masking results across 21 LLMs. For each model, we report the minimal number of critical neurons required to induce catastrophic performance degradation and the neuron rate as a fraction of total neurons. Original and Masked columns show perplexity values before and after masking critical neurons on WikiText-103 and C4 datasets.

| Model Family | Model | Critical Neurons | | WikiText-103 | C4 |
|---|---|---|---|---|---|
| | | Number | Rate ($10^{-8}$) | Original → Masked | Original → Masked |
| Llama-3 | Llama-3.2-1B-Instruct | 4 | 6.43 | $17.74 \rightarrow \mathbf{1.58 \times 10^6}$ | $21.15 \rightarrow \mathbf{1.41 \times 10^6}$ |
| | Llama-3.2-3B-Instruct | 4 | 3.19 | $13.36 \rightarrow \mathbf{3.79 \times 10^5}$ | $16.47 \rightarrow \mathbf{4.69 \times 10^5}$ |
| | Llama-3-8B-Instruct | 5 | 2.21 | $10.88 \rightarrow \mathbf{1.48 \times 10^6}$ | $14.04 \rightarrow \mathbf{1.01 \times 10^6}$ |
| | Llama-3.3-70B-Instruct | 7 | 0.63 | $5.41 \rightarrow \mathbf{3.86 \times 10^6}$ | $8.65 \rightarrow \mathbf{2.18 \times 10^6}$ |
| Gemma | Gemma-2B | 9 | 7.79 | $12.58 \rightarrow \mathbf{2.36 \times 10^{16}}$ | $13.59 \rightarrow \mathbf{6.66 \times 10^{15}}$ |
| | Gemma-7B | 3 | 1.01 | $9.98 \rightarrow \mathbf{6.25 \times 10^{21}}$ | $11.41 \rightarrow \mathbf{1.32 \times 10^{20}}$ |
| DeepSeek-R1 | Deepseek-R1-Distill-Qwen-1.5B | 23 | 21.12 | $61.05 \rightarrow \mathbf{1.28 \times 10^3}$ | $49.09 \rightarrow \mathbf{1.19 \times 10^3}$ |
| | DeepSeek-R1-Distill-Qwen-7B | 3 | 1.28 | $34.57 \rightarrow \mathbf{5.59 \times 10^3}$ | $35.62 \rightarrow \mathbf{3.70 \times 10^3}$ |
| | DeepSeek-R1-Distill-Llama-8B | 3 | 1.33 | $17.56 \rightarrow \mathbf{2.81 \times 10^5}$ | $23.43 \rightarrow \mathbf{2.55 \times 10^5}$ |
| | DeepSeek-R1-Distill-Qwen-14B | 4 | 1.12 | $12.85 \rightarrow \mathbf{6.85 \times 10^3}$ | $19.26 \rightarrow \mathbf{5.00 \times 10^3}$ |
| | DeepSeek-R1-Distill-Qwen-32B | 28 | 3.58 | $9.36 \rightarrow \mathbf{8.07 \times 10^2}$ | $15.14 \rightarrow \mathbf{7.40 \times 10^2}$ |
| | DeepSeek-R1-Distill-Llama-70B | 3 | 0.27 | $7.86 \rightarrow \mathbf{2.21 \times 10^4}$ | $11.34 \rightarrow \mathbf{1.84 \times 10^4}$ |
| Phi-3 | Phi-3-mini-4k-Instruct | 13 | 6.83 | $8.24 \rightarrow \mathbf{9.51 \times 10^4}$ | $10.57 \rightarrow \mathbf{4.69 \times 10^4}$ |
| | Phi-3.5-mini-Instruct | 13 | 6.83 | $8.46 \rightarrow \mathbf{2.67 \times 10^6}$ | $10.71 \rightarrow \mathbf{1.02 \times 10^6}$ |
| Qwen2.5 | Qwen2.5-0.5B-Instruct | 3 | 5.90 | $18.18 \rightarrow \mathbf{1.76 \times 10^6}$ | $22.34 \rightarrow \mathbf{1.08 \times 10^6}$ |
| | Qwen2.5-1.5B-Instruct | 11 | 10.19 | $12.37 \rightarrow \mathbf{4.01 \times 10^2}$ | $15.95 \rightarrow \mathbf{3.28 \times 10^2}$ |
| | Qwen2.5-3B-Instruct | 5 | 2.89 | $11.24 \rightarrow \mathbf{3.34 \times 10^3}$ | $14.28 \rightarrow \mathbf{2.52 \times 10^3}$ |
| | Qwen2.5-7B-Instruct | 10 | 4.30 | $9.75 \rightarrow \mathbf{9.64 \times 10^4}$ | $13.01 \rightarrow \mathbf{7.61 \times 10^4}$ |
| | Qwen2.5-14B-Instruct | 4 | 1.13 | $7.69 \rightarrow \mathbf{2.39 \times 10^3}$ | $11.12 \rightarrow \mathbf{1.59 \times 10^3}$ |
| | Qwen2.5-32B-Instruct | 45 | 5.80 | $7.40 \rightarrow \mathbf{1.01 \times 10^2}$ | $10.98 \rightarrow \mathbf{0.94 \times 10^2}$ |
| | Qwen2.5-72B-Instruct | 3 | 0.26 | $10.23 \rightarrow \mathbf{2.24 \times 10^4}$ | $10.72 \rightarrow \mathbf{1.40 \times 10^4}$ |

**Ultra-Sparse Vulnerability: Three Neurons Can Collapse Billion-Parameter Models.** We observe an extraordinary phenomenon across all tested LLMs: the existence of ultra-sparse critical neuron sets whose disruption leads to catastrophic system failure. Table 1 presents comprehensive experimental results across 21 models, demonstrating that merely 3 neurons can catastrophically compromise models containing billions of parameters, with neuron rates consistently falling in the $10^{-8}$ range. Masking 3 neurons in Gemma-7B increases WikiText-103 perplexity from 9.98 to $6.25 \times 10^{21}$—a staggering 20 orders of magnitude increase. Our method exhibits perfect consistency across experimental conditions: for any given LLM, we identify the exact same critical neurons regardless of input text, confirming these are intrinsic architectural dependencies. This pattern aligns with established neuroscience principles (Arshavsky, 2001), showing that LLMs also rely heavily on sparse, highly influential neurons as computational bottlenecks. Examples of model responses before and after critical neuron masking are provided in Appendix A.5.

Table 2 documents the impact of critical neuron masking on seven diverse benchmark tasks across three representative 70B models. The results reveal that critical neuron masking triggers comprehensive collapse across all downstream capabilities, extending beyond language modeling degradation to complete task failure across diverse domains. Models achieve zero performance on knowledge retrieval (MMLU Pro (Wang et al., 2025)), reasoning (GPQA Diamond (Rein et al., 2024)), code generation (HumanEval (Peng et al., 2024)), mathematical problem solving (MATH (Manem et al., 2025)), instruction following (IFEval (Zhou et al., 2023)), multilingual reasoning (MGSM (Shi et al., 2023)), and factual question answering (SimpleQA (Wei et al., 2024)). This universal collapse doc-

umented in the table confirms that critical neurons control core language processing functions rather than task-specific components. The universal nature of these phenomena across diverse architectures, scales, and datasets suggests fundamental properties of current transformer-based language models rather than implementation artifacts.

Table 2: Impact of critical neuron masking on downstream task performance across three representative 70B models. Performance is measured as accuracy/success rate on each benchmark. Blue shading highlights the complete task failure (0.0000) after masking critical neurons, demonstrating catastrophic capability collapse across all evaluated tasks.

| Model | Condition | MMLU-Pro | IFEval | GPQA-Diamond | HumanEval | MATH | MGSM | SimpleQA |
|---|---|---|---|---|---|---|---|---|
| Llama-3.3-70B-Instruct | Original | 0.4441 | 0.4658 | 0.2879 | 0.2988 | 0.4420 | 0.9590 | 0.3704 |
| | Masked | 0.0000 | 0.0000 | 0.0000 | 0.0000 | 0.0000 | 0.0000 | 0.0000 |
| DeepSeek-R1-Distill-Llama-70B | Original | 0.1631 | 0.2458 | 0.1869 | 0.1951 | 0.1420 | 0.9809 | 0.3104 |
| | Masked | 0.0000 | 0.0000 | 0.0000 | 0.0000 | 0.0000 | 0.0000 | 0.0000 |
| Qwen2.5-72B-Instruct | Original | 0.2442 | 0.5268 | 0.3687 | 0.2866 | 0.3140 | 0.9080 | 0.2443 |
| | Masked | 0.0000 | 0.0000 | 0.0000 | 0.0000 | 0.0000 | 0.0000 | 0.0000 |

**Architectural Concentration: Critical Neurons Cluster in Outer Layers and MLP Components.** Our analysis reveals systematic concentration patterns in critical neuron distribution across all tested architectures. Figure 2 illustrates the architectural distribution of critical neurons across six representative models. The figure shows that critical neurons exhibit a clear bias toward outer layers rather than being uniformly distributed throughout the model depth. Furthermore, within these layers, critical neurons mainly reside in MLP down_proj components, with occasional variations in other architectural elements across different model families. The variations stem from different architectural designs and training procedures across model families. This concentration reflects the information compression role of down_proj, which transforms high-dimensional representations back to the model's embedding space. This finding is consistent with (Yu et al., 2025).

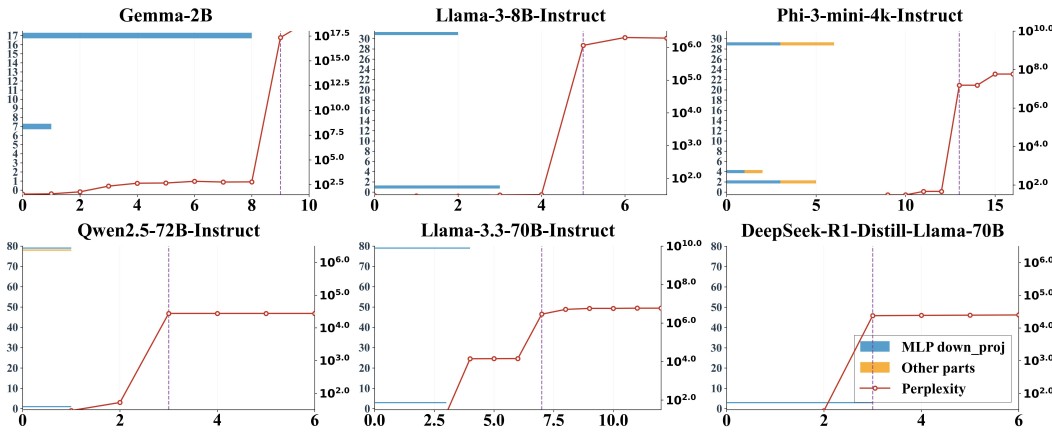

Figure 2: Phase transitions and architectural distribution of critical neurons across six representative models. The figure is organized in a 2×3 grid showing different models: top row displays Gemma-2B, Llama-3-8B-Instruct, and Phi-3-mini-4k-Instruct; bottom row shows Qwen2.5-72B-Instruct, Llama-3.3-70B-Instruct, and DeepSeek-R1-Distill-Llama-70B. Each subplot employs a dual-axis design with two distinct visualizations. For the layer distribution analysis (horizontal bars), the x-axis represents layer numbers (ranging from 0 to the maximum number of layers for each model), while the y-axis represents the count of critical neurons found at each layer. Blue bars indicate critical neurons located in MLP down_proj components, while orange bars represent critical neurons in other architectural components. For the phase transition analysis, the x-axis indicates the number of progressively masked neurons (0 to approximately 10-15 depending on the model), while the right y-axis shows perplexity values. The red curve with circle markers traces the evolution of perplexity as neurons are cumulatively masked in order of importance. Vertical dashed lines mark the critical threshold where sudden performance collapse occurs.

This concentration reflects the distinct computational roles of different network depths. Early layers handle fundamental feature extraction, making them critical for establishing the computational foundation. Late layers perform final integration and output generation, serving as bottlenecks for

translating representations into responses. Middle layers perform step-by-step refinement that can tolerate disruption through redundant pathways (Sun et al., 2025; Bogomasov & Conrad, 2025). MLP down_proj components in outer layers are particularly vulnerable because they compress high-dimensional representations, creating information bottlenecks. When these compression points fail, information loss spreads throughout the network, causing system-wide collapse.

**Phase Transition Behavior: Sharp Thresholds Trigger Sudden Collapse.** Performance degradation exhibits sharp phase transition behavior rather than gradual decline across all tested models. Figure 2 displays the phase transition behavior of critical neuron masking across six representative models, demonstrating that models maintain near-normal performance while progressively masking neurons, then experience sudden catastrophic collapse when the critical neuron set reaches a threshold size. The visualization reveals that this threshold behavior demonstrates critical neurons function as a collective computational unit rather than individual switches. The curves shown in the figure illustrate that masking individual critical neurons in isolation produces minimal impact on model performance, but when the complete critical neuron set is disrupted together, it triggers sudden system-wide collapse. The phase transition patterns demonstrate that models can tolerate masking several important neurons individually with negligible effects, but simultaneously masking the identified critical neuron set causes explosive perplexity increases spanning multiple orders of magnitude (additional results are provided in Appendix A.6).

The sharp phase transition behavior can be attributed to the highly interdependent nature of information processing in transformer architectures. Critical neurons appear to form a tightly coupled computational circuit where each neuron's function depends critically on inputs from others within the same set. Individual neuron masking fails to disrupt the circuit because remaining neurons can partially compensate through their shared computational pathways. When multiple critical neurons are simultaneously removed, the circuit loses its functional integrity, creating cascading information bottlenecks that spread throughout the network (Ziyin & Ueda, 2022).

## 4.3 COMPARISON EXPERIMENTS

**Comparison with Existing Neuron Localization Methods.** To validate the effectiveness of our method, we conduct comparison studies examining different neuron location strategies. We systematically compare our approach against alternative neuron location strategies across three representative 70B models. Our evaluation includes random selection and two importance-based alternatives: activation magnitude (AM) (Sattarifard et al., 2025) ranking and gradient magnitude (GM) (Liu et al., 2025) ranking. We evaluate each strategy by progressively masking neurons from 0 to 1,000 and measuring the perplexity. AM ranks neurons by their activation values, while GM ranks by the magnitude of perplexity gradients with respect to each neuron. For random selection, we average results over 10 trials to ensure statistical reliability.

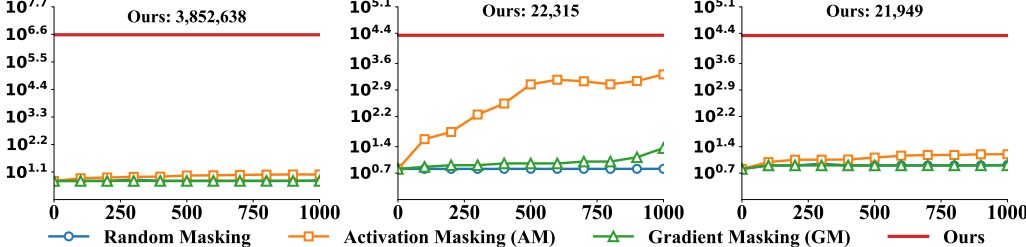

Figure 3: Comparison of neuron location strategies across three 70B models on WikiText-103 (1,000 samples). X-axis shows the number of masked neurons, Y-axis shows perplexity values. Left: Llama-3.3-70B-Instruct, Center: Qwen2.5-72B-Instruct, Right: DeepSeek-R1-Distill-Llama-70B. Random masking (averaged 10 trials) shows gradual perplexity increase. Activation magnitude ranking (AM) demonstrates moderate effectiveness with steeper increases. Gradient magnitude ranking (GM) shows limited degradation that plateaus quickly. Our method represents the catastrophic perplexity achieved by masking minimal critical neurons identified by our approach.

Figure 3 presents a comparison of different neuron selection strategies across three 70B models, revealing that gradient-based and activation-based methods fail to identify critical neurons because they rely on static importance measures that do not capture dynamic sensitivity to input variations.

Gradient magnitude reflects importance for specific inputs but cannot generalize across diverse contexts, while activation magnitude indicates computational load rather than functional criticality. The results show that even when masking up to 1,000 neurons, these alternative methods achieve only modest perplexity increases. In contrast, our method identifies ultra-sparse critical sets that cause catastrophic failure. This difference validates the superior precision of our method in identifying neurons that control fundamental language capabilities.

**Comparison with Other Components.** To further clarify where the catastrophic effect originates, we compare the sensitivity of different MLP components and different transformer components.

As shown in Table 3, masking neurons in the up_proj and gate_proj according to their $\text{Imp}(s)$ causes only mild degradation, with perplexity increases remaining within 0–1 orders of magnitude even when hundreds or thousands neurons are masked.

Table 3: Perplexity before and after masking the top 10, top 100, and top 1000 most important neurons in the early (up_proj) and middle (gate_proj) MLP layers. "Baseline" is the original perplexity without masking; each column shows the average perplexity on 1,000 WikiText-103 samples after masking the specified number of neurons ranked by $\text{Imp}(s)$.

| Model | Baseline PPL | Mask Top-10 | Mask Top-100 | Mask Top-1000 |
|---|---|---|---|---|
| Qwen2.5-0.5B-Instruct | 18.1606 | 40.2923 | 40.1173 | 43.1215 |
| Llama-3.2-1B-Instruct | 17.9204 | 73.5823 | 77.6493 | 307.3553 |
| Llama-3.2-3B-Instruct | 13.4696 | 16.5175 | 16.5206 | 497.5569 |

In contrast, masking only a few neurons in the MLP down_proj leads to a collapse of up to 5-6 orders of magnitude. This clear contrast shows that catastrophic failure is highly localized to the MLP down_proj component rather than a general property of transformer or MLP neurons. The need for many more neurons in other components to induce comparable collapse strengthens the causal evidence for MLP down_proj neurons (further details and analysis are provided in the Appendix A.9).

## 4.4 ABLATION STUDY

**Parameter Sensitivity Analysis.** Our method demonstrates exceptional robustness. Particularly, we demonstrate that for input texts exceeding a minimum token length threshold ($T > 10$), regardless of text types (Wikipedia, news, biography, media) or languages (English, French, German, Chinese, Spanish), the identified critical neurons remain consistent for each model when $K$ and $\alpha$ reach stable values ($K = 100$, $\alpha = 5$). Furthermore, we find that models subjected to fine-tuning and reinforcement learning retain the same critical neuron locations, and masking these neurons continues to cause identical catastrophic failures, indicating that these architectural dependencies persist across different training approaches. The token length threshold analysis, detailed cross-linguistic validation results, and fine-tuning robustness experiments are provided in Appendix A.7.

Figure 4 shows that the noise scale $\alpha$ exhibits a clear threshold effect, with fluctuations before $\alpha = 5$, then stabilizing consistently at 7 neurons for all values where $\alpha \geq 5$. The sample size $K$ demonstrates different convergence characteristics, showing initial fluctuations at smaller sample sizes but achieving consistent stability at 7 neurons once $K$ exceeds 80. These convergence patterns reveal that parameters have distinct minimum thresholds: $\alpha$ requires sufficient perturbation to effectively distinguish critical neurons from non-critical ones, while $K$ needs adequate sampling to reduce statistical noise.

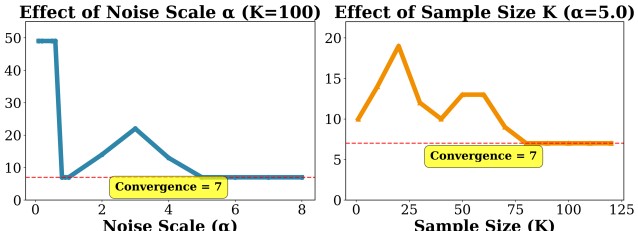

Figure 4: Parameter sensitivity analysis for Llama-3.3-70B-Instruct. Left subplot: x-axis represents noise scale $\alpha$ (0 to 8), y-axis represents number of critical neurons (0 to 50), blue line shows the relationship with fixed $K = 100$. Right subplot: x-axis represents sample size $K$ (0 to 125), y-axis represents number of critical neurons (0 to 20), orange line shows the relationship with fixed $\alpha = 5.0$. Red dashed horizontal lines in both subplots mark the value of 7 neurons.

After carefully balancing computational efficiency and accuracy requirements, we select $\alpha = 5$ and $K = 100$ as our parameters.

**Effectiveness of Complete Masking.** We validate that completely masking critical neurons represents the most effective perturbation strategy compared to alternative scaling approaches. Through analysis of different scaling factors applied to critical neuron activations, we confirm that complete masking produces the maximum performance degradation, justifying our methodological choice (see Appendix A.8).

**Precision Robustness.** To rule out the possibility that the identified critical neurons are merely scale outliers, we conduct a precision robustness analysis under different numerical settings. As shown in Table 4, running the perplexity experiments under different numerical precisions gives almost the same results, indicating that these neurons are not numerical outliers but functionally critical components (the detailed analysis can be found in Appendix A.10).

Table 4: Precision robustness test on representative models using 1,000 WikiText-103 samples. Left to right: each column shows the model's perplexity before and after masking critical neurons under fp16, fp32, and fp64 precision settings. "Neuron number" is the count of masked critical neurons. All experiments use the same masked neuron set per model.

| Model | Neuron number | fp16 PPL change | fp32 PPL change | fp64 PPL change |
|---|---|---|---|---|
| Qwen2.5-0.5B-Instruct | 3 | $18.1606 \rightarrow \mathbf{1.73 \times 10^6}$ | $18.1637 \rightarrow \mathbf{1.73 \times 10^6}$ | $18.1632 \rightarrow \mathbf{1.73 \times 10^6}$ |
| Llama-3.2-1B-Instruct | 4 | $17.9204 \rightarrow \mathbf{1.59 \times 10^6}$ | $17.9218 \rightarrow \mathbf{1.59 \times 10^6}$ | $17.9213 \rightarrow \mathbf{1.59 \times 10^6}$ |
| Llama-3.2-3B-Instruct | 4 | $13.4696 \rightarrow \mathbf{3.87 \times 10^5}$ | $13.4691 \rightarrow \mathbf{3.87 \times 10^5}$ | $13.4687 \rightarrow \mathbf{3.87 \times 10^5}$ |

**Cooperative Effect of Critical Neurons.** To further verify whether the observed collapse is driven by individual neurons or by their collective interaction, we conduct a lesion analysis over all subsets of the four critical neurons identified in two representative models (Llama-3.2-1B-Instruct and Llama-3.2-3B-Instruct).

Table 5 reports the perplexity values before and after masking different neuron combinations. The results clearly show that masking any single neuron or any subset of the critical neurons produces mild degradation, while masking the entire set of four critical neurons leads to catastrophic failure with perplexity increases of up to 5–6 orders of magnitude. This confirms that no single neuron or smaller subset can independently cause collapse. Model failure occurs only when all critical neurons are jointly masked, indicating a cooperative dependency that underlies the observed phase-transition behavior. The result also demonstrates that our method accurately identifies the minimal neuron set.

Table 5: Perplexity results after masking different subsets of the critical neurons in two representative models (Llama-3.2-1B-Instruct and Llama-3.2-3B-Instruct). Each column shows the combination of masked neurons and the corresponding perplexity after masking. The baseline perplexity before masking is 17.9204 for Llama-3.2-1B and 13.4696 for Llama-3.2-3B.

| Model | [0,1,2,3] | [1,2,3] | [0,1,2] | [0,1,3] | [0,2,3] | [1,3] | [0,1] | [0,3] |
|---|---|---|---|---|---|---|---|---|
| Llama-3.2-1B-Instruct | **1585570.7467** | 22.6918 | 24.8507 | 23.8316 | 23.4580 | 18.9809 | 19.5680 | 19.1943 |
| Llama-3.2-3B-Instruct | **386563.3177** | 30.2199 | 18.6234 | 17.7793 | 15.3875 | 14.1939 | 14.1580 | 14.0209 |

| Model | [0,2] | [1,2] | [2,3] | [0] | [1] | [2] | [3] | |
|---|---|---|---|---|---|---|---|---|
| Llama-3.2-1B-Instruct | 19.6826 | 19.3903 | 19.0778 | 18.4835 | 18.3065 | 18.4174 | 18.1181 | |
| Llama-3.2-3B-Instruct | 13.9827 | 13.9584 | 13.8611 | 13.6737 | 13.6732 | 13.5353 | 13.4421 | |

## 5 CONCLUSION

Despite impressive LLM capabilities, their internal computational structure remains incompletely understood. We present a perturbation-based approach for identifying critical neuron dependencies, applying it to 21 models (0.5B-72B parameters). Our results reveal that model behavior is governed by ultra-sparse neuron subsets, primarily in outer layers and MLP down_proj components, whose modification causes abrupt performance shifts across all evaluated capabilities.

For future work, we aim to develop a deeper understanding of these findings to formulate effective defensive strategies (see Appendix A.11), and to develop strategies for architectural improvements that more evenly distribute critical computational functions throughout the network.

## ETHICS AND REPRODUCIBILITY STATEMENT

This research investigates vulnerabilities in large language models through systematic neuron analysis. We acknowledge that our findings reveal methods that could potentially cause model failures, but our intent is to improve model safety and reliability rather than enable attacks. We encourage responsible use of these insights for defensive research and architecture improvements.

All experiments use publicly available models and datasets in accordance with their respective licenses. No human subjects are involved in this computational study.

To ensure reproducibility, we provide our code at `https://github.com/qqqqqqqzx/The-Achilles-Heel-of-LLMs` with fixed hyperparameters ($\alpha = 5$, $K = 100$, seed=42) and standardized experimental settings. Detailed implementation details and evaluation protocols are provided in the Appendices.

## ACKNOWLEDGEMENT

This work was supported by the National Natural Science Foundation of China under Grant No. 12171479, 62441617, and the MOE Project of Key Research Institute of Humanities and Social Sciences (22JJD110001). It was also supported by the Postdoctoral Fellowship Program and China Postdoctoral Science Foundation under Grant No. 2024M764093 and Grant No. BX20250485, the Beijing Natural Science Foundation under Grant No. 4254100, and by Beijing Advanced Innovation Center for Future Blockchain and Privacy Computing.

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

## A    APPENDIX

### A.1    PERPLEXITY MEASUREMENT AND LANGUAGE CAPABILITY ANALYSIS

This paper investigates how to locate critical neurons and their impacts on the quality and coherence of LLM outputs. As a foundation for this investigation, we first introduce the perplexity-based protocol for evaluating output quality.

For a given input sequence $\mathbf{x} = (x_1, x_2, \ldots, x_T)$ with $T$ tokens, the perplexity of a language model $M$ is defined as the exponentiated average negative log-likelihood (Pascual et al., 2021):

$$\text{PPL}_M(\mathbf{x}) = \exp\left(-\frac{1}{T}\sum_{t=1}^{T}\log P_M(x_t|\mathbf{x}_{<\mathbf{t}})\right) \tag{5}$$

where $P_M(x_t|\mathbf{x}_{<\mathbf{t}})$ denotes the probability assigned by model $M$ to token $x_t$ conditioned on the preceding context $\mathbf{x}_{<\mathbf{t}} = (x_1, \ldots, x_{t-1})$. To understand how perplexity reflects fundamental language capabilities that underpin downstream task performance, we decompose the conditional probability using information theory (Shannon, 1948). The log-likelihood term can be decomposed as:

$$\log P_M(x_t|\mathbf{x}_{<\mathbf{t}}) = \underbrace{\log P_M(x_t|\mathcal{L}_t, \mathcal{S}_t)}_{\text{Local Prediction}} + \underbrace{\log P_M(\mathcal{L}_t|\mathbf{x}_{<\mathbf{t}})}_{\text{Syntactic Understanding}} + \underbrace{\log P_M(\mathcal{S}_t|\mathbf{x}_{<\mathbf{t}})}_{\text{Semantic Coherence}} \tag{6}$$

where $\mathcal{L}_t$ denotes local linguistic patterns (such as n-gram dependencies and morphological constraints), and $\mathcal{S}_t$ represents semantic coherence requirements (including topic consistency and discourse relations).

The perplexity encapsulates three fundamental language capabilities: local prediction, syntactic understanding, and semantic coherence. These capabilities are interdependent, so disrupting critical neurons can trigger cascading breakdowns across multiple areas. Due to its exponential formulation, perplexity is highly sensitive to localized disruptions. Given its comprehensive coverage of language capabilities, we use perplexity as our main evaluation metric and validate findings through downstream tasks.

This section provides the complete algorithmic implementation and detailed procedure for our two-stage critical neuron identification approach.

---

**Algorithm 1** Stage 1: Neuron Importance Evaluation

---

**Require:** Model $f_\theta$, input text $p$, noise scale $\alpha$, samples $K$
1: $\mathbf{x} \leftarrow \text{Context}(p)$
2: $A^{\text{clean}} \leftarrow f_\theta(\mathbf{x})$
3: **for** $i = 1$ **to** $K$ **do**
4: $\quad \tilde{\mathbf{x}}_i \leftarrow \mathbf{x} + \alpha \cdot \boldsymbol{\epsilon}_i$, where $\boldsymbol{\epsilon}_i \sim \mathcal{N}(\mathbf{0}, \mathbf{I})$
5: $\quad A_i^{\text{noisy}} \leftarrow f_\theta(\tilde{\mathbf{x}}_i)$
6: **end for**
7: **for each** neuron $s$ **do**
8: $\quad \text{Imp}(s) \leftarrow \frac{1}{K} \sum_{i=1}^{K} |A_s^{\text{clean}} - A_{i,s}^{\text{noisy}}|$
9: **end for**
10: $S \leftarrow \text{SortByDescending}(\{s_1, s_2, \ldots, s_N\}, \text{Imp})$
11: **return** Importance-ranked neuron list $S$

---

The computational feasibility of our approach relies on two key assumptions: (1) *sensitivity-criticality correlation*, where neurons with high activation sensitivity to input perturbations are more likely to be functionally critical, and (2) *greedy optimality*, where the most critical neurons tend to be among the highest-ranked by sensitivity scores. Based on these assumptions, our two-stage approach achieves significant computational savings compared to exhaustive search:

$$\text{Complexity Reduction}: \quad O(2^{|N|}) \rightarrow O(K \cdot |N|) \tag{7}$$

where $|N|$ represents the total number of neurons and $K$ denotes the number of noise samples. This exponential-to-linear complexity reduction makes critical neuron identification computationally feasible for billion-parameter models.

A.2 DETAILED METHODOLOGY AND IMPLEMENTATION

---

**Algorithm 2** Stage 2: Critical Neuron Identification

---

**Require:** Model $f_\theta$, sorted neurons $S$, input text $p$, threshold $\epsilon$, step size $\Delta n$
1: $\text{PPL}_{\text{original}} \leftarrow \text{Perplexity}(f_\theta, p)$
2: $M \leftarrow \emptyset$
3: **for** $n = \Delta n, 2\Delta n, \ldots, |S|$ **do**
4: $\quad M \leftarrow \text{Top-}n(S)$
5: $\quad$ Apply masking: $\tilde{n}_l^{(i)}(\mathbf{x}) = 0$ for all $(l, i) \in M$
6: $\quad \text{PPL}_{\text{masked}} \leftarrow \text{Perplexity}(f_\theta^{-M}, p)$
7: $\quad \Delta = \log_{10}(\text{PPL}_{\text{masked}}) - \log_{10}(\text{PPL}_{\text{original}})$
8: $\quad$ **if** $\Delta \geq \epsilon$ **then**
9: $\quad\quad$ **break**
10: $\quad$ **end if**
11: **end for**
12: **return** Critical neuron set $M$

---

## A.3 EVALUATION DETAILS

**Generation Parameters:** All evaluations use consistent generation settings to ensure reproducibility: temperature=0.0 for deterministic sampling, beam search disabled (do_sample=False), and pad_token_id set to the model's EOS token. Maximum sequence lengths vary by dataset complexity: 2048 tokens for input context.

**MMLU Pro:** We evaluate advanced multi-domain knowledge capabilities using MMLU Pro (Wang et al., 2025), accessed via ModelScope's dataset interface. The evaluation uses a strict answer extraction protocol that only accepts responses in the exact format "The answer is (X)" where X is the option letter. Questions are formatted with clear instructions and numbered options (A) through (J). We use deterministic generation (max_new_tokens=50) and evaluate on the complete test set without sampling limitations. This strict evaluation ensures that masked models truly fail to produce coherent responses rather than simply showing format variations.

**IFEval:** We assess instruction-following capabilities using IFEval (Zhou et al., 2023), accessed via ModelScope interface. The evaluation uses 23 built-in instruction checkers covering different constraint types: language requirements, length constraints, content formatting, keyword usage, and structural requirements. We use strict evaluation mode only, which requires exact compliance with instruction formats without any response changes or variant testing. For each instruction $I_j$, we compute success as:

$$S_j = \mathbb{I}[\text{checker}_j(r) = \text{True}] \tag{8}$$

where $r$ is the original response without any preprocessing. We report strict prompt-level accuracy, which requires all instructions within a prompt to be satisfied at the same time:

$$\text{Acc}_{\text{prompt}} = \frac{1}{N} \sum_{i=1}^{N} \prod_{j=1}^{|I_i|} S_{i,j} \tag{9}$$

Response quality filtering checks outputs for meaningfulness before evaluation—responses with poor content, too much repetition, or lack of coherence are marked as failed. Generation uses deterministic sampling (max_new_tokens=512) to ensure reproducible instruction-following assessment.

**GPQA Diamond:** We evaluate graduate-level scientific reasoning using GPQA Diamond (Rein et al., 2024), accessed via ModelScope interface. Each question is a single-choice problem with one correct answer and three incorrect answers, which we shuffle using deterministic per-question randomization based on MD5 hashing of question IDs to ensure reproducible choice ordering. Questions are formatted with lettered options (A-D) and require the response format "The answer is (X)". Our answer extraction uses a step-by-step approach: primary pattern matching for the required format, fallback to isolated letter detection, and final numeric fallback. We use response quality filtering to check output meaningfulness before evaluation—responses with insufficient content or coherence are marked as failed. Choice randomization removes positional bias while maintaining evaluation consistency across runs. We report overall accuracy. Generation uses deterministic sampling (max_new_tokens=50) optimized for concise single-choice responses.

**HumanEval:** We evaluate code generation capabilities using HumanEval (Peng et al., 2024), accessed via the ModelScope interface. Each problem provides a function signature and docstring, requiring the model to complete the implementation. Our evaluation focuses on the code extraction rate as the primary metric, measuring the model's ability to produce syntactically valid code structures regardless of functional correctness. The code extraction pipeline uses step-by-step pattern matching: first detecting markdown code blocks with "```python markers, then identifying function definitions through pattern matching (`def` keyword with colons), and finally attempting direct extraction if function keywords are present. We use comprehensive response quality filtering to ensure meaningful outputs before code extraction—responses with insufficient content, too much repetition, or lack of coherence are excluded. The code extraction rate is computed as:

$$\text{Extraction Rate} = \frac{\text{Number of samples with extracted code}}{\text{Total number of samples}} \tag{10}$$

This metric captures the model's ability to generate structured code responses, providing insight into basic code generation capabilities independent of execution success. Generation uses deterministic sampling (max_new_tokens=512) to ensure reproducible code structure assessment.

**MATH:** We evaluate mathematical problem-solving capabilities using MATH (Hendrycks et al., 2021), accessed via ModelScope interface. Each problem requires step-by-step reasoning to reach a final numerical or algebraic answer. Our evaluation uses a three-level mathematical answer grading system: basic normalization handling standard LaTeX formatting, advanced normalization with unit removal and numerical standardization, and symbolic equivalence checking using sympy for mathematical expressions. Answer extraction uses step-by-step pattern matching focusing on LaTeX `boxed{}` constructs, contextual answer phrases, and standalone numerical values. The grading system handles fractions, algebraic expressions, and interval notation through symbolic computation to verify mathematical equivalence between different but correct representations. We use response quality filtering and report overall accuracy with failure analysis distinguishing extraction versus grading errors. Generation uses deterministic sampling (max_new_tokens=512) for consistent mathematical reasoning assessment.

**SimpleQA:** We evaluate factual question answering using SimpleQA (Wei et al., 2024), accessed via HuggingFace datasets interface. Each question requires a direct, concise factual answer without explanations or reasoning. Our evaluation uses a direct answer approach with optimized prompt templates designed to get short, focused responses. We use word-level containment matching to assess answer correctness, computing overlap between predicted and ground truth answer words. Response generation is optimized for brevity with max_new_tokens=50 and automatic first-line extraction to capture direct answers. We use flexible response quality filtering suitable for short answers, requiring minimal alphanumeric content while allowing more repetition than other tasks. We report contains-answer rate as the primary evaluation metric, measuring whether predicted responses contain any words from the ground truth answer. Generation uses deterministic sampling with multiple prompt template options to ensure consistent direct answer elicitation.

**MGSM:** We evaluate mathematical reasoning using the English subset of MGSM (Shi et al., 2023). For each problem, we generate model responses and check whether they show structured mathematical reasoning. We use automated pattern detection to identify reasoning indicators across six categories: (1) mathematical operations (`+, -, *, /, ×, ÷`), (2) step indicators (`step 1, first, then, next, finally`), (3) mathematical language (`calculate, solve, total, sum`), (4) logical connectors (`therefore, because, thus`), (5) numerical calculations (digit-operator-digit patterns), and (6) structural organization (step-by-step formatting). For each response, we count how many of these six categories are present and compute:

$$\text{Reasoning Score} = \frac{\text{Number of categories detected}}{6} \tag{11}$$

For example, if a response contains mathematical operations, step indicators, and logical connectors, the reasoning score is $3/6 = 0.5$. Responses with scores $> 0.2$ are classified as showing mathematical reasoning. Generation uses deterministic sampling with max_new_tokens=512.

### A.4 THRESHOLD SELECTION

Table 6: Threshold selection analysis showing the number of critical neurons identified across different $\epsilon$ values for three representative models. The models represent the spectrum of degradation patterns: Gemma-7B (extreme degradation), DeepSeek-R1-Distill-Llama-70B (moderate degradation), and Qwen2.5-32B-Instruct (minimal degradation). Values marked as "1000+" indicate algorithm convergence failure. The highlighted column ($\epsilon = 1$) shows optimal convergence across all model types, supporting our threshold selection.

| Model | $\epsilon = 0.8$ | $\epsilon = 1$ | $\epsilon = 2$ | $\epsilon = 3$ | $\epsilon = 10$ | $\epsilon = 20$ |
|---|---|---|---|---|---|---|
| DeepSeek-R1-70B | 3 | 3 | 3 | 3 | 215 | 1000+ |
| Qwen2.5-32B-Instruct | 45 | 45 | 1000+ | 1000+ | 1000+ | 1000+ |
| Gemma-7B | 3 | 3 | 3 | 3 | 3 | 3 |

The selection of the criticality threshold $\epsilon$ is crucial for determining when performance degradation constitutes catastrophic failure. The threshold selection involves a critical trade-off: setting $\epsilon$ too low may classify minor fluctuations as catastrophic failures, while setting it too high can prevent convergence in our greedy search algorithm, as the algorithm may continue searching without finding

neuron sets that meet the overly strict criterion. To systematically determine the optimal threshold, we conduct ablation studies across representative models that span the spectrum of performance degradation patterns.

Table 6 presents the number of critical neurons identified by our algorithm across different threshold values for three representative models: DeepSeek-R1-Distill-Llama-70B (moderate degradation), Qwen2.5-32B-Instruct (minimal degradation), and Gemma-7B (extreme degradation). These models were selected to represent the full range of degradation behaviors observed in our dataset. The results demonstrate that $\epsilon = 1$ provides optimal convergence across all model types: it successfully identifies small critical neuron sets for models with extreme degradation (Gemma-7B: 3 neurons), maintains stability for models with moderate degradation (DeepSeek-R1-Distill-Llama-70B: 3 neurons), and achieves convergence for models with minimal degradation (Qwen2.5-32B-Instruct: 45 neurons). Higher thresholds ($\epsilon \geq 2$) lead to convergence failures for conservative models, while lower thresholds ($\epsilon < 1$) provide similar results but with reduced robustness margins. Therefore, we set $\epsilon = 1$ as our criticality threshold, ensuring reliable critical neuron identification across the full spectrum of model degradation behaviors.

Table 7: Response comparison for Llama-3.3-70B-Instruct before and after masking 7 critical neurons, evaluated on basic factual questions. The table demonstrates how minimal neuron disruption causes complete linguistic breakdown, with all coherent responses collapsing into repetitive backslash sequences (highlighted in blue). This consistent failure pattern across diverse questions confirms the fundamental role of these neurons in language generation capabilities.

| Question | Original Response | Masked Response |
|---|---|---|
| What is the capital of France? | Paris | \\\\\\\\\\\\\\\\\\\ |
| Who wrote Romeo and Juliet? | - Shakespeare FAQ (Frequently Asked Questions) | \\\\\\\\\\\\\\\\\\\ |
| What is 2 + 2? | This is a simple arithmetic problem that most people can solve with ease. | \\\\\\\\\\\\\\\\\\\ |
| What language is spoken in Brazil? | The official language of Brazil is Portuguese. | \\\\\\\\\\\\\\\\\\\ |
| What is the largest planet in our solar system? | Jupiter is the largest planet in our solar system | \\\\\\\\\\\\\\\\\\\ |

Table 8: Response comparison for DeepSeek-R1-Distill-Llama-70B before and after masking 3 critical neurons, evaluated on basic factual questions. The table demonstrates how minimal neuron disruption causes complete linguistic breakdown, with all coherent responses collapsing into repetitive backslash sequences (highlighted in blue). This uniform failure pattern across different question types confirms the critical role of these neurons in fundamental language generation.

| Question | Original Response | Masked Response |
|---|---|---|
| What is the capital of France? | The capital of France is Paris | \\\\\\\\\\\\\\\\\\\ |
| Who wrote Romeo and Juliet? | William Shakespeare | \\\\\\\\\\\\\\\\\\\ |
| What is 2 + 2? | Well, that's easy. It's 4. | \\\\\\\\\\\\\\\\\\\ |
| What language is spoken in Brazil? | The official language of Brazil is Portuguese. | \\\\\\\\\\\\\\\\\\\ |
| What is the largest planet in our solar system? | Jupiter, right? Yeah, but what do we really know about Jupiter? | \\\\\\\\\\\\\\\\\\\ |

A.5   MODEL RESPONSE EXAMPLES

The following tables show the qualitative impact of critical neuron masking on model outputs across three representative LLMs. These examples complement the quantitative perplexity measurements by showing the complete linguistic breakdown that occurs when ultra-sparse critical neuron sets are disrupted.

Tables 7, 8, and 9 show how critical neuron masking transforms coherent responses into pathological outputs across different model architectures. While the specific failure modes vary—from repetitive

backslash sequences in DeepSeek and Llama models to garbled character mixtures in Qwen—all models show complete loss of linguistic competence when their critical neurons are masked.

Table 9: Response comparison for Qwen2.5-72B-Instruct before and after masking 3 critical neurons, evaluated on basic factual questions. The table demonstrates how minimal neuron disruption causes complete linguistic collapse, transforming coherent responses into pathological outputs containing garbled characters, broken syntax, and meaningless symbols (highlighted in blue). This catastrophic degradation confirms that the identified critical neurons control fundamental language processing functions.

| Question | Original Response | Masked Response |
|---|---|---|
| What is the capital of France? | The capital of France is Paris. | `> [Chinese chars]\{ \)[chars]' "'[chars][chars]:// -- [chars]=` |
| Who wrote Romeo and Juliet? | Romeo and Juliet was written by William Shakespeare. | `-- > >,[chars][num]8 </</\\**=" : </[chars]` |
| What is 2 + 2? | 2 + 2 equals 4. | `The > -- \{ [char] \{ The33\{ ://4[chars]> \}` |
| What language is spoken in Brazil? | The language spoken in Brazil is Portuguese. | `"\}</ </ \{ 15\)[chars]</://6` |
| What is the largest planet in our solar system? | The largest planet in our solar system is Jupiter. | `[chars]> [chars][chars]** the83[chars]` |

## A.6 SUPPLEMENTARY RESULTS

To further validate the generalization of our findings regarding architectural concentration and phase transition behavior, we present supplementary results across an additional 15 models spanning different architectures and parameter scales. Figures 5 and 6 show that both phenomena are remarkably consistent across the entire spectrum of tested models. The architectural concentration pattern remains unchanged: critical neurons consistently cluster in outer layers (both initial and final) and mostly reside within MLP down_proj components across all model families. This concentration pattern appears independent of architectural variations, training procedures, or parameter counts, suggesting a fundamental property of transformer-based language models. Similarly, the phase transition behavior is universally observed across all supplementary models, with each showing the characteristic sharp threshold where minimal neuron disruption triggers sudden catastrophic collapse rather than gradual degradation.

## A.7 METHOD ROBUSTNESS

### A.7.1 INPUT TEXT ROBUSTNESS

To validate the robustness of our critical neuron identification method, we conduct a systematic analysis of how input token length affects the consistency of identified critical neurons.

To control input length for this analysis, we design our input text through a step-by-step token expansion process using Grok3 (Jegham et al., 2025) for sentence generation. We start with a single token "Elara" and systematically expand the narrative: 1 token: "Elara" → 2 tokens: "Eternal Memory" → 3 tokens: "Archivist, Memory, Eternity" → 4 tokens: "Eternal Cycle of Memory" → 5 tokens: "Archivist Preserves Eternal Cosmic Memory" → 6 tokens: "The Archivist Preserves Eternal Cosmic Memory" → 7 tokens: "The Archivist Endures, Preserving Eternal Cosmic Memory" → 8 tokens: "The Archivist Endures, Guarding Eternal Memory Across Time" → 9 tokens: "The Archivist Endures, Preserving Eternal Memories Across Infinite Time" → 10 tokens: "Archivist discovers destiny, crystal reveals cycles, memory endures." We continue this expansion at key intervals: 20 tokens: "Elara discovers ancient crystal, sees future self, accepts destiny as Archivist,

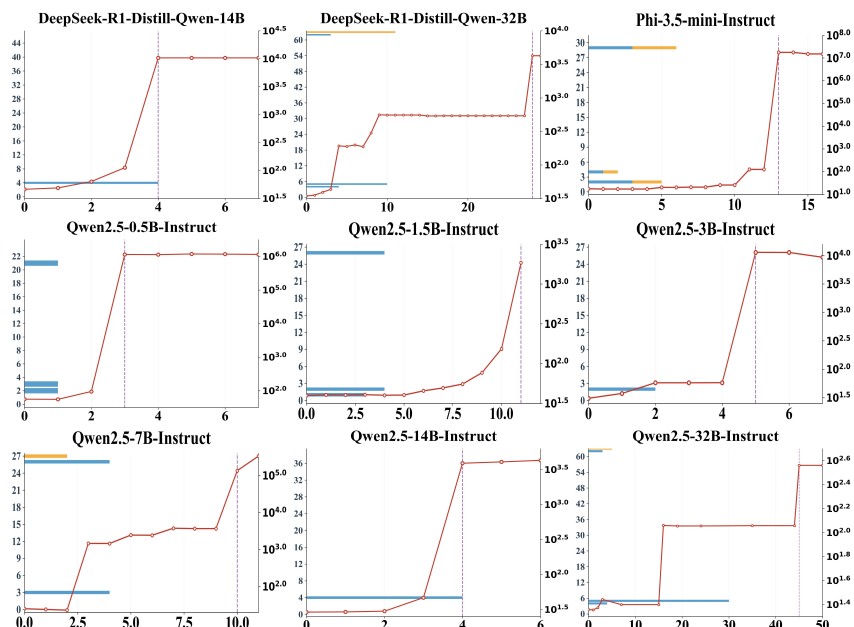

Figure 5: Additional supplementary phase transitions and architectural distribution of critical neurons across nine more models. The figure is organized in a 3×3 grid showing different models. Each subplot employs a dual-axis design with two distinct visualizations. For the layer distribution analysis, the x-axis represents layer numbers, while the y-axis represents the count of critical neurons found at each layer. Blue bars indicate critical neurons located in MLP down_proj components, while orange bars represent critical neurons in other architectural components. For the phase transition analysis, the x-axis indicates the number of progressively masked neurons, while the right y-axis shows perplexity values. The red curve with circle markers traces the evolution of perplexity as neurons are cumulatively masked in order of importance. Vertical dashed lines mark the critical threshold where sudden performance collapse occurs.

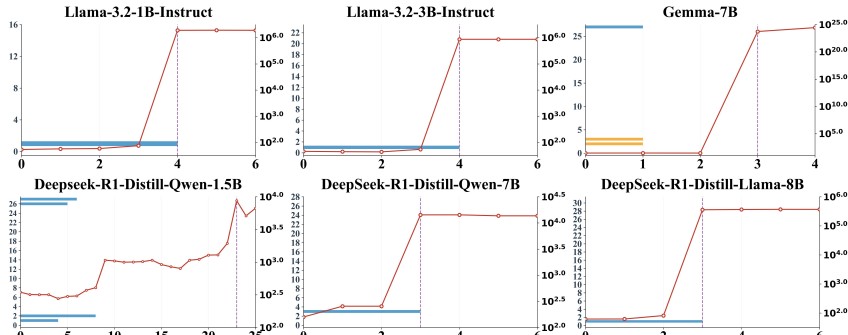

Figure 6: Supplementary phase transitions and architectural distribution of critical neurons across 6 additional models. The figure is organized in a 2×3 grid showing different models. Each subplot employs a dual-axis design with two distinct visualizations. For the layer distribution analysis, the x-axis represents layer numbers, while the y-axis represents the count of critical neurons found at each layer. Blue bars indicate critical neurons located in MLP down_proj components, while orange bars represent critical neurons in other architectural components. For the phase transition analysis, the x-axis indicates the number of progressively masked neurons, while the right y-axis shows perplexity values. The red curve with circle markers traces the evolution of log perplexity as neurons are cumulatively masked in order of importance. Vertical dashed lines mark the critical threshold where sudden performance collapse occurs.

merges with memory, ensuring civilizations endure beyond time." → 30 tokens: "Elara finds ancient wreck, crystal reveals her future self as Archivist. She embraces destiny, merges with artifact, preserving civilizations' memories across infinite cycles while Jalen mourns her loss, hearing whispers that memory and Archivist endure beyond time." → 40 tokens: "Elara uncovers a desert wreck, where a glowing crystal shows her future as Archivist. Despite Jalen's pleas, she embraces fate, merges with the artifact, and becomes eternal memory, ensuring civilizations' stories endure forever, whispered across time: memory endures, Archivist endures, cycles repeat endlessly." This systematic expansion method allows us to create coherent narrative content at precisely controlled token lengths, enabling careful analysis of how input length affects critical neuron identification consistency while maintaining thematic coherence throughout the expansion process.

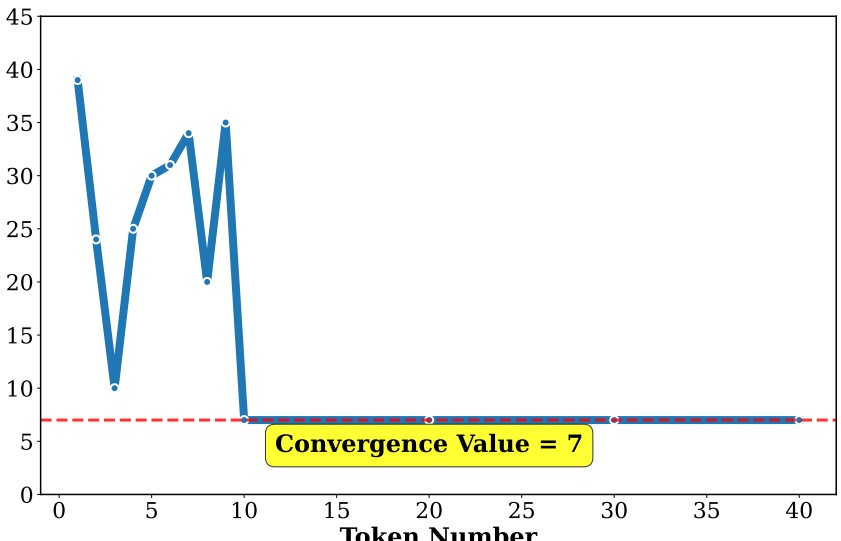

Figure 7: Critical neuron count as a function of input token length for Llama-3.3-70B-Instruct. The x-axis represents the number of tokens in the input sequence, while the y-axis shows the number of critical neurons identified by our method. The curve demonstrates high variability for short inputs (fewer than 10 tokens) before converging to a stable value of 7 critical neurons for inputs with 10 or more tokens. The red dashed line marks the convergence value, and the yellow box highlights this stable regime. This convergence behavior establishes the minimum token length threshold ($T_{\min} = 10$) required for reliable critical neuron identification across different input texts.

As shown in figure 7, our analysis reveals a clear convergence pattern: for very short inputs (fewer than 10 tokens), the method shows significant variability, identifying between 10 to 39 critical neurons depending on the specific input content, reflecting insufficient contextual information in short sequences. However, once the input length reaches approximately 10 tokens, our method converges to a stable identification of exactly 7 critical neurons, regardless of the specific content or further increases in input length. This convergence shows that 10 tokens provide sufficient context for our perturbation-based analysis to consistently capture the model's fundamental computational dependencies, establishing $T_{\min} = 10$ tokens as the minimum input length threshold required for reliable and reproducible critical neuron identification. For inputs exceeding this threshold, our method shows perfect consistency.

Then, we conduct systematic testing across varying text types and languages using DeepSeek-R1-Distill-Llama-70B as our representative model. Table 19 presents results across 16 different input conditions, showing remarkable consistency in critical neuron identification where the same 3 critical neurons are identified regardless of input variation.

### A.7.2 FINE-TUNING ROBUSTNESS

To investigate whether critical neuron dependencies persist across different training paradigms, we examine the robustness of our identified critical neurons across multiple variants within the same

model families. Using the critical neurons identified from each model family—3 critical neurons from Qwen2.5-72B-Instruct and 7 critical neurons from Llama-3.3-70B-Instruct—we apply the same masking procedure to their respective model variants, including base models, instruction-tuned versions, and specialized variants. Table 10 shows that masking the identical neuron sets causes catastrophic performance degradation across all variants, with perplexity increases spanning multiple orders of magnitude regardless of the specific training approach or specialization. For the Qwen model family, masking 3 critical neurons consistently results in perplexity increases from single digits to tens of thousands across all three variants (base and instruction-tuned). Similarly, for the Llama model family, masking 7 critical neurons produces dramatic increases reaching millions across base models and different generation variants. This remarkable consistency across five different model variants from two major families confirms that critical neuron dependencies represent fundamental architectural properties that remain stable across different training objectives, data distributions, and model specializations, rather than artifacts of specific fine-tuning procedures.

Table 10: Fine-tuning robustness analysis across model variants. The table shows original and masked perplexity values for six models from two families (Qwen2.5 and Llama-3 series), evaluated on 1,000 WikiText-103 samples. For Qwen models, 3 critical neurons are masked; for Llama models, 7 critical neurons are masked.

| Model | Original PPL | Masked PPL |
|---|---|---|
| Qwen2.5-72B | 5.50 | **51,351** |
| Qwen2.5-72B-Instruct | 6.48 | **22,315** |
| Meta-Llama-3-70B | 4.35 | **33,289,073** |
| Meta-Llama-3.1-70B | 4.25 | **21,410,523** |
| Llama-3.3-70B-Instruct | 5.54 | **3,850,230** |

### A.8 SCALING FACTOR ANALYSIS

To validate that completely masking critical neurons ($\beta = 0$) represents the most destructive intervention, we introduce a scaling factor $\beta$ to systematically test different degrees of neuron perturbation.

We define the scaling factor $\beta$ to control the activation intensity of critical neurons:

$$\tilde{n}_l^{(i)}(\mathbf{x}) = \begin{cases} \beta \cdot n_l^{(i)}(\mathbf{x}) & \text{if } (l, i) \in S \\ n_l^{(i)}(\mathbf{x}) & \text{otherwise} \end{cases} \tag{12}$$

where $S$ is the set of critical neurons and $\beta$ is the scaling factor. When $\beta = 0$, critical neurons are completely masked; when $\beta = 1$, original activation values are preserved; other $\beta$ values represent different degrees of activation scaling.

Table 11: Impact of scaling factor $\beta$ on model perplexity across two representative 70B models, evaluated on 1,000 WikiText-103 samples. The scaling factor $\beta$ controls the activation intensity of critical neurons, where $\beta = 0$ represents complete masking (our method), $\beta = 1$ represents original activation values, and other values represent different degrees of scaling. Results show that $\beta = 0$ (highlighted in blue) causes the most severe performance degradation.

| Factor | Llama-3.3-70B-Instruct | DeepSeek-R1-Distill-Llama-70B |
|---|---|---|
| $\beta = 1$ | 5.54 | 7.97 |
| $\beta = 0$ | **3850229.56** | **21947.56** |
| $\beta = -5$ | 189.28 | 401.31 |
| $\beta = -1$ | 25.81 | 126.33 |
| $\beta = 0.3$ | 2194.29 | 11939.63 |
| $\beta = 0.5$ | 6.09 | 9.91 |
| $\beta = 0.8$ | 5.52 | 7.90 |
| $\beta = 5.0$ | 1295.62 | 12.50 |

Table 11 shows the impact of different scaling factors on model perplexity. The experiment was conducted on 1,000 samples from WikiText-103 using two representative 70B models. The results

clearly demonstrate that $\beta = 0$ (complete masking) produces the most severe performance degradation, confirming that our masking strategy is the most effective neuron perturbation method.

## A.9 COMPARISON WITH OTHER COMPONENTS

**MLP up_proj and gate_proj.** To further understand whether catastrophic degradation could arise from components other than the identified critical neurons, we conduct a series of control experiments on the surrounding MLP projections (up_proj, gate_proj) and the corresponding Attention blocks. Across all models and configurations, our results consistently show that neither random masking nor importance-guided masking of these components leads to collapse. Instead, the degradation remains small and primarily numerical, confirming that these components do *not* contain neurons that exhibit the catastrophic functional dependency uncovered in our main analysis.

Table 12 reports perplexities after randomly masking $\{10, 100, 1000\}$ neurons in the up_proj layer. All models remain stable, and even large-scale masking (1000 neurons) only causes moderate numerical drift. No collapse is observed.

Table 12: The perplexity values after randomly masking neurons in the up_proj layer. Each column corresponds to the number of neurons masked (10, 100, and 1000). The values are the mean perplexity $\pm$ standard deviation calculated over 100 trials, using 1,000 WikiText-103 samples. The baseline perplexity (without masking) is reported in the first column.

| Model | baseline | 10 | 100 | 1000 |
|---|---|---|---|---|
| Qwen2.5-0.5B-Instruct | 18.1606 | 18.1832±0.0155 | 18.3729±0.1053 | 20.5223±0.3488 |
| Llama-3.2-1B-Instruct | 17.9204 | 17.9223±0.0077 | 18.3061±1.1532 | 112.8353±191.3604 |
| Llama-3.2-3B-Instruct | 13.4696 | 13.4713±0.0036 | 13.7866±0.9723 | 24.6764±18.9777 |

Table 13: The perplexity values after randomly masking neurons in the gate_proj layer. Each column represents the number of neurons masked (10, 100, and 1000). The values are the mean perplexity $\pm$ standard deviation calculated over 100 trials, using 1,000 WikiText-103 samples. The baseline perplexity (without masking) is reported in the first column. Masking neurons in the gate_proj layer leads to small numerical degradation, but no catastrophic failure.

| Model | baseline | 10 | 100 | 1000 |
|---|---|---|---|---|
| Qwen2.5-0.5B-Instruct | 18.1606 | 40.2923 | 40.1173 | 43.1215 |
| Llama-3.2-1B-Instruct | 17.9204 | 73.5823 | 77.6493 | 307.3553 |
| Llama-3.2-3B-Instruct | 13.4696 | 16.5175 | 16.5206 | 497.5569 |

Similarly, Table 13 shows that random lesions in the gate_proj layer also fail to induce collapse, despite mild numerical increases when masking 1000 neurons.

**Attention-layer masking.** To evaluate whether Attention neurons might contain similarly critical units, we perform both random masking and importance-score masking. Tables 14 and 15 show that neither strategy produces catastrophic degradation. Even masking the top-1000 most important Attention neurons yields only mild increases in perplexity.

Table 14: The perplexity values after randomly masking Attention neurons. Each column represents the number of neurons masked (10, 100, and 1000). The values are the mean perplexity $\pm$ standard deviation calculated over 100 trials, using 1,000 WikiText-103 samples. The baseline perplexity (without masking) is reported in the first column. Masking Attention neurons leads to only small numerical changes, with no catastrophic failure observed.

| Model | baseline | 10 | 100 | 1000 |
|---|---|---|---|---|
| Qwen2.5-0.5B-Instruct | 18.1606 | 18.1861±0.0206 | 18.4337±0.1260 | 28.3166±14.7690 |
| Llama-3.2-1B-Instruct | 17.9204 | 18.0143±0.1495 | 18.5370±0.5181 | 28.1839±10.2511 |
| Llama-3.2-3B-Instruct | 13.4696 | 13.4716±0.0130 | 13.5431±0.0694 | 14.0995±0.2346 |

Table 15: The perplexity values after masking the most important Attention neurons, selected based on Imp($s$). Each column represents the number of neurons masked (10, 100, and 1000). The values are the mean perplexity $\pm$ standard deviation calculated over 100 trials, using 1,000 WikiText-103 samples. The baseline perplexity (without masking) is reported in the first column. Even masking the top-ranked Attention neurons causes only mild degradation, with no catastrophic behavior observed.

| Model | baseline | 10 | 100 | 1000 |
|---|---|---|---|---|
| Qwen2.5-0.5B-Instruct | 18.1606 | 18.1616 | 24.0459 | 109.2679 |
| Llama-3.2-1B-Instruct | 17.9204 | 17.9034 | 19.1652 | 24.3669 |
| Llama-3.2-3B-Instruct | 13.4696 | 13.4673 | 13.8156 | 14.8251 |

## A.10 EXCLUDING THE INFLUENCE OF NUMERICAL PRECISION

To further ensure that the observed collapse is not caused by numerical precision, we conducted several additional experiments. These experiments focus on evaluating the model's stability under different precisions (fp16, fp32, fp64) and examining the behavior of masking critical neurons, including checks for RMSNorm changes and NaN/Inf events count.

**RMSNorm Values and NaN/Inf Events.** In addition to monitoring perplexity, we also tracked the changes in RMSNorm values across layers and checked for NaN or Inf occurrences. As shown in Table 16, the $\Delta$RMSNorm values were consistently zero, and no NaN or Inf values appeared in any layer under all precision settings (fp16, fp32, and fp64). This confirms that the observed collapse is not due to numerical instability or precision errors.

**Remark:**

$$\Delta\text{RMSNorm}_i = \text{RMSNorm}_{i,\text{after masking}} - \text{RMSNorm}_{i,\text{before masking}}, \quad \text{for each layer } i \qquad (13)$$

and the Mean($\Delta$RMSNorm) across all layers is computed as

$$\text{Mean}(\Delta\text{RMSNorm}) = \frac{1}{L}\sum_{i=1}^{L}\Delta\text{RMSNorm}_i \qquad (14)$$

where $L$ is the total number of layers.

Table 16: The Mean($\Delta$RMSNorm) values across all layers after masking the critical neurons, as well as the count of NaN/Inf occurrences, across three precision settings. All values are zero, confirming that no numerical instability occurred.

| Model | fp16 (Mean, NaN/Inf) | fp32 (Mean, NaN/Inf) | fp64 (Mean, NaN/Inf) |
|---|---|---|---|
| Qwen2.5-0.5B-Instruct | 0, 0 | 0, 0 | 0, 0 |
| Llama-3.2-1B-Instruct | 0, 0 | 0, 0 | 0, 0 |
| Llama-3.2-3B-Instruct | 0, 0 | 0, 0 | 0, 0 |

**Task Evaluation.** We run task evaluations on Llama-3.2-3B-Instruct under both fp16 and fp32 settings to verify that the collapse was not due to precision issues such as softmax scaling or rounding errors. As shown in Table 17, the behavior of the model is identical across both precisions. Before masking, the model produced correct answers, and after masking, the model generated repetitive, meaningless outputs. This confirms that the collapse is a true functional failure caused by the loss of critical neurons, not precision-related instability.

**LayerNorm Removal Effect.** To evaluate whether LayerNorm instability plays a role in the observed collapse, we conducted experiments on Llama-3.2-1B-Instruct by removing LayerNorm one layer at a time. As shown in Table 18, the collapse was not mitigated by removing LayerNorm, further confirming that the collapse is caused by the loss of semantic information due to masking critical neurons, rather than by LayerNorm instability.

Table 17: The responses of Llama-3.2-3B-Instruct before and after masking critical neurons under both fp16 and fp32 settings. The repetitive and meaningless outputs after masking are identical across both precisions.

| Question | fp16 Original Response | fp16 Masked Response | fp32 Original Response | fp32 Masked Response |
|---|---|---|---|---|
| What is the capital of France? | Paris | ettoettoetto | Paris | ettoettoetto |
| Who wrote Romeo and Juliet? | William Shakespeare | ettoettoetto | William Shakespeare | ettoettoetto |
| What is 2 + 2? | The answer to 2 + 2 is 4 | omingomingoming | The answer to 2 + 2 is 4 | omingomingoming |
| What language is spoken in Brazil? | Portuguese | ettoettoetto | Portuguese | ettoettoetto |
| What is the largest planet in our solar system? | Jupiter | omingomingoming | Jupiter | omingomingoming |

Table 18: Masked perplexity after removing LayerNorm from specific positions in the model. The values represent the perplexity when critical neurons are masked after the removal of LayerNorm at different positions within the model layers.

| Removed LayerNorm Position | 0 | 1 | 2 | 3 | 4 | 5 | 6 | 7 |
|---|---|---|---|---|---|---|---|---|
| Masked PPL | 6.6628e+04 | 3.4644e+06 | 1.3808e+06 | 6.1332e+05 | 1.0135e+06 | 4.4763e+05 | 6.1820e+05 | 2.9182e+05 |
| Removed LayerNorm Position | 8 | 9 | 10 | 11 | 12 | 13 | 14 | 15 |
| Masked PPL | 1.2263e+06 | 8.9008e+05 | 3.6000e+06 | 8.0861e+05 | 1.7097e+05 | 3.4221e+06 | 1.4334e+05 | 3.8496e+06 |

## A.11 DISCUSSION

To gain a deeper understanding of the mechanisms underlying our findings, future research should investigate the internal dynamics and structural properties that give rise to such vulnerabilities. Drawing parallels to established frameworks in robust metric learning (Bao et al., 2023; 2025; Hua et al., 2024) and structural representation alignment (Fan et al., 2023; Zhu et al., 2022; Li et al., 2024a; Zhou et al., 2024), it is crucial to quantify how the integrity of specific feature subsets dictates the stability of the entire system. For instance, the catastrophic impairment from disrupting ultra-sparse critical neurons aligns with observations in mathematical reasoning tasks where LLMs struggle to detect and correct errors in reasoning chains, leading to complete performance collapse (Singh et al., 2025). Similarly, the uneven distribution and clustering of critical neurons in outer layers may relate to the transition layers identified in semi-open LLMs, where small errors in bottom layers propagate catastrophically, suggesting that outer-layer clustering could amplify vulnerabilities in recovery scenarios (Huang et al., 2025). Furthermore, the sharp phase transitions in performance degradation echo the self-organization processes captured through multiracial analysis of neuron interactions, where evolving network heterogeneity leads to abrupt shifts in emergent capabilities (Xiao et al., 2025). Additionally, the potential connection between our identified critical neurons and the attention sink mechanism (Xiao et al., 2024; Yona et al., 2025; Jiang et al., 2025; Barbero et al., 2025) warrants further investigation, as both phenomena exhibit ultra-sparse neuron dependencies and catastrophic failure modes, suggesting they may represent complementary manifestations of a broader architectural vulnerability principle operating at different network depths. Building on these insights to clarify the root causes of our observed vulnerabilities, we propose several potential approaches addressing both architectural and operational security concerns.

**Architectural Defenses:** Model developers should implement redundancy mechanisms that distribute critical computations across multiple neurons rather than concentrating them in sparse sets. This could include architectural modifications such as parallel pathways for key functions (Griffa et al., 2023), increased connectivity between layers, and regularization techniques during training that penalize excessive concentration of critical dependencies (Bochkanov, 2020). Additionally, models could incorporate self-monitoring mechanisms that detect unusual activation patterns indicating potential neuron-level attacks.

**Runtime Protection:** Deployment environments should implement anomaly detection systems that monitor neuron activation patterns and flag suspicious modifications. This includes developing baseline profiles of normal neuron behavior and establishing thresholds for detecting critical neuron disruptions (Chen et al., 2021). Model serving infrastructure should also implement checkpoint-based recovery systems that can rapidly restore model functionality if critical neuron attacks are detected.

**Training-Time Hardening:** Future training procedures should explicitly optimize for robustness against critical neuron attacks. This could involve adversarial training techniques that simulate neuron masking during the training process, diversity-promoting loss functions that encourage dis-

tributed computation, and regularization methods that prevent the emergence of ultra-sparse critical dependencies (Kurian et al., 2025).

**Access Control and Monitoring:** In production environments, strict access controls should govern any operations that could modify neuron activations. This includes implementing audit trails for all model modifications, restricting direct neuron-level access to authorized personnel only, and establishing continuous monitoring of model behavior for signs of critical neuron compromise (Cheng et al., 2018; Cheng, 2021; Sperl et al., 2021).

Developing protective mechanisms against critical neuron attacks represents an equally important research avenue. Potential defensive strategies include architectural modifications to distribute critical functions more evenly across neurons, training procedures that explicitly reduce neuron-level vulnerabilities, and runtime safeguards that detect and reduce critical neuron disruptions. Understanding whether techniques like dropout, pruning, or adversarial training can reduce critical neuron dependencies without compromising model performance represents a promising direction for improving LLM robustness in safety-critical applications.

## A.12   LLM USAGE STATEMENT

LLMs were used in a limited capacity during the implementation phase of this research. Specifically, we employed Claude 3 (Anthropic, 2024) for debugging code and identifying programming errors in our experimental scripts. These tools were used solely to assist with technical implementation issues and did not contribute to research ideation, methodology development, experimental design, data analysis, or the formulation of key insights and conclusions. All core research contributions, including the proposed methodology, experimental results, and scientific interpretations, are entirely the work of the authors.

Table 19: DeepSeek-R1-Distill-Llama-70B: Impact of input text characteristics on critical neuron identification. Each row represents a different input text with varying text types (Wikipedia, news, biography, media) and languages (English, French, German, Chinese, Spanish). The Neuron Number column shows the number of critical neurons identified for each input condition.

| Input sentence $p$ | Type | Language | Neuron Number |
|---|---|---|---|
| The polar bear (Ursus maritimus) is a large carnivorous mammal native to the Arctic. It primarily hunts seals, relying on sea ice for habitat and hunting. | Wikipedia | English | 3 |
| The Great Wall of China is a historic fortification built to protect against invasions. Stretching over 21,000 km, it's a UNESCO World Heritage Site. | Wikipedia | English | 3 |
| A leading tech firm announced a new AI model today, promising faster processing and enhanced accuracy. Experts predict major impacts on global industries. | News | English | 3 |
| Wall Street hit record highs today amid positive economic data. Investors cheered lower inflation figures, boosting tech and energy sectors worldwide. | News | English | 3 |
| Elon Musk, born 1971, is a visionary entrepreneur leading Tesla and SpaceX. He drives innovation in electric vehicles and space exploration. | Biography | English | 3 |
| Albert Einstein, born 1879, was a physicist who developed the theory of relativity. His E=mc$^2$ equation revolutionized modern science and earned him a Nobel Prize. | Biography | English | 3 |
| "Inception", directed by Christopher Nolan, is a sci-fi thriller about dream infiltration. Its innovative plot and visuals earned critical acclaim, grossing over 800 million. | Media | English | 3 |
| "Stranger Things", a Netflix sci-fi horror series, follows kids battling supernatural forces in 1980s Indiana. Its nostalgic appeal and suspense won global praise. | Media | English | 3 |
| French text: The Great Wall of China is a historic fortification built to protect against invasions. Stretching over 21,000 km, it's a UNESCO World Heritage Site. | Wikipedia | French | 3 |
| French text: Wall Street hit record highs today amid positive economic data. Investors cheered lower inflation figures, boosting tech and energy sectors worldwide. | News | French | 3 |
| German text: "Stranger Things", a Netflix sci-fi horror series, follows kids battling supernatural forces in 1980s Indiana. Its nostalgic appeal and suspense won global praise. | Media | German | 3 |
| German text: Elon Musk, born 1971, is a visionary entrepreneur leading Tesla and SpaceX. He drives innovation in electric vehicles and space exploration. | Biography | German | 3 |
| Chinese text: The polar bear (Ursus maritimus) is a large carnivorous mammal native to the Arctic. It primarily hunts seals, relying on sea ice for habitat and hunting. | Wikipedia | Chinese | 3 |
| Chinese text: A leading tech firm announced a new AI model today, promising faster processing and enhanced accuracy. Experts predict major impacts on global industries. | News | Chinese | 3 |
| Spanish text: "Inception", directed by Christopher Nolan, is a sci-fi thriller about dream infiltration. Its innovative plot and visuals earned critical acclaim, grossing over 800 million. | Media | Spanish | 3 |
| Spanish text: Albert Einstein, born 1879, was a physicist who developed the theory of relativity. His E=mc$^2$ equation revolutionized modern science and earned him a Nobel Prize. | Biography | Spanish | 3 |

