# OpenReview forum: "The Achilles’ Heel of LLMs: How Altering a Handful of Neurons Can Cripple Language Abilities"
_ICLR.cc/2026/Conference — ICLR 2026 Poster_

### Official Review · Reviewer_SYxg · 2025-10-26

**Soundness:** 4
**Presentation:** 3
**Contribution:** 1
**Rating:** 2
**Confidence:** 4

**Summary:**

This paper finds that LLMs contain a sparse set of critical neurons whose disruption can cause catastrophic performance failure. The authors propose a perturbation-based method to identify these neurons and evaluate it across multiple models. They report three main findings: (1) disrupting even three neurons can collapse models with billions of parameters, (2) critical neurons concentrate in outer layers and MLP down_proj components, and (3) performance degradation exhibits sharp phase transitions rather than gradual decline.

**Strengths:**

1. Comprehensive experimental scope: The evaluation spans 21 models across multiple architectures (Llama, Gemma, DeepSeek, Phi, Qwen) and diverse benchmarks, providing strong empirical evidence for the phenomena.

2. Clear methodology: The two-stage approach (sensitivity analysis + causal verification) is clearly presented with algorithmic details.

3. Robust validation: The paper includes thorough ablation studies examining parameter sensitivity (α, K), input length requirements (T>10), cross-lingual consistency, and fine-tuning robustness.

**Weaknesses:**

1. Limited Novelty of Core Findings
The paper acknowledges that concentration in MLP down_proj layers "is consistent with the observations reported in Yu et al. (2025)". The existence of "super weights" in these components is already documented. What fundamentally new insight does this work provide beyond confirming known vulnerabilities at the neuron level rather than weight level?

**Recommendation:** I would encourage to distinguish more clearly how this work advances beyond Yu et al. (2025) on "The Super Weight in Large Language Models"

2. Insufficient Theoretical Depth
While the empirical observations are extensively documented across many models, the paper lacks sufficient explanation of the underlying mechanisms. The phenomenon is thoroughly characterized in terms of what happens (phase transitions, concentration patterns, catastrophic failure), but the *why* and *how* remain underexplored.

**Recommendation:** To strengthen the contribution, I recommend providing deeper theoretical analysis grounded in optimization dynamics, information theory, or circuit analysis to explain why these vulnerabilities emerge and what they fundamentally reveal about transformer computation.

3. Unclear Practical Implications
The paper frames its findings as identifying vulnerabilities with implications for "deployment security in safety-critical applications" (line 22), yet the practical actionability of these findings remains unclear. Two critical questions are unaddressed: (1) How can practitioners suppress or mitigate the emergence of these critical neurons during training or deployment? (2) What is the realistic threat model—who would have the capability to selectively mask individual neurons in production LLMs, and under what circumstances? While Section A.9 discusses potential defenses, it reads as speculative rather than validated. The gap between identifying the vulnerability and demonstrating exploitability weakens the security framing.

**Recommendation:** To strengthen practical impact, clarify the threat model with concrete attack scenarios, and either demonstrate mitigation strategies empirically or reframe the contribution as primarily advancing scientific understanding rather than addressing immediate security concerns.

4. Overextended Neuroscience Analogy
The neuroscience motivation (Section 1, lines 6-30) establishes useful context by drawing parallels to sparse criticality in biological neural networks. However, the analogy is invoked repeatedly throughout the paper without deepening the connection beyond surface-level metaphor. This creates an expectation for substantive interdisciplinary dialogue—for example, demonstrating whether LLM critical neurons exhibit functional properties analogous to biological critical neurons, or whether insights from neuroscience can inform mitigation strategies. As currently presented, the neuroscience framing risks appearing cosmetic rather than integral to the contribution.

**Recommendation:** Either (1) reduce the emphasis on neuroscience to a brief motivational context in the introduction, or (2) strengthen the interdisciplinary connection by establishing concrete links—such as comparing your findings to specific neuroscience phenomena (e.g., do critical neurons in LLMs exhibit analogous plasticity, recovery patterns, or functional specialization documented in biological systems?).

**Questions:**

Connection to Related Phenomena: Attention Sinks
Your findings bear notable similarity to prior work on attention sinks, which identified highly important neurons critical for model fluency. Specifically, Xiao et al. (2023) documented the attention sink phenomenon (https://arxiv.org/abs/2309.17453), which was subsequently traced to highly activated neurons in specific architectural locations (https://arxiv.org/pdf/2503.08908). This mechanistic understanding has enabled cleaner interpretability analyses (https://arxiv.org/abs/2506.08010) and deeper theoretical investigation of the underlying dynamics (https://arxiv.org/abs/2504.02732).
Questions for the authors:

Is there a connection between the critical neurons you identify and the attention sink mechanism? Given that both phenomena involve sparse, highly influential neurons whose disruption causes catastrophic failure, investigating potential overlap could be illuminating.
If a relationship exists, the extensive theoretical framework developed for attention sinks (particularly https://arxiv.org/abs/2504.02732) might provide a foundation for explaining why your critical neurons emerge and how they function. This could address the theoretical depth concern raised above.

---

> ### Author Response · Authors · 2025-11-19
>
> We sincerely thank you for your detailed review and thoughtful feedback. We appreciate your recognition of the reproducibility of our method, the clarity of our two-stage approach, and the breadth of our experimental evaluation across 21 models. In the following response, we address each of your points one by one.

---

> ### Author Response · Authors · 2025-11-19
> **Part 1/6**
>
> ## Weakness 1
>
> Most of reviewers have noted the need to clarify the distinction between our work and *Yu et al.* (2025) [1].
>
> We have clarified this distinction in the revised ***Related Work* (red font)** section. While both studies examine how a very small subset of parameters or units can disproportionately affect overall model performance, they differ fundamentally in **research objectives, disruption magnitude, methodological design, mechanism discovery, and the content and process of the conclusions**:
>
> 1. Disruption strength and effect level: *Yu et al.* (2025) analyzes individual **weights**, where perturbations increase perplexity by up to 3 orders of magnitude and only reduce task accuracy to random guessing. Our work targets **neuron-level lesions**, where masking just 3–7 neurons increases perplexity by up to **20 orders of magnitude** and entirely destroys task understanding. This reveals a deeper **functional collapse** rather than a numerical perturbation.
> 2. Methodological scalability and engineering applicability: The method of *Yu et al.* (2025) relies on manual observation to identify super weights, limiting reproducibility. Our proposed **perturbation–causal verification framework** automatically and systematically locates critical neurons, forming a standardized, reproducible pipeline that can be directly applied in engineering practice to identify and protect these critical neurons.
> 3. Discovery of multi-neuron interaction mechanisms: We find that collapse is not triggered by a single “super neuron,” but by the **collective interaction of a small group of critical neurons**, Any subset alone fails to cause comparable damage (see Ablation Study purple font), exhibiting a **phase-transition** phenomenon. *Yu et al.* (2025) does not explore such multi-unit interaction or nonlinear catastrophic mechanisms.
> 4. Different research units: *Yu et al.* (2025) investigates **parameter-level** importance, reflecting primarily numerical amplification effects. In contrast, our analysis operates at the **neuron level**, uncovering the causal and functional roles of neurons and identifying structural bottlenecks underlying high-level language abilities.
> 5. Differences in conclusions and discovery process: Rather than focusing on numerical outliers, our findings emerge from analyzing functional collapse patterns, which reveal sharp phase transitions when coordinated neuron groups are disrupted.
> 6. Much broader experimental coverage and generalization: Our findings are supported by a far broader and more systematic experimental scope, covering 21 models from 0.5B to 72B and 10 datasets, which provides stronger generalization than the settings studied in *Yu et al.* (2025).
>
>
>
> [1]Mengxia Yu, De Wang, Qi Shan, Colorado J Reed, Alvin Wan   **The Super Weight in Large Language Models**     https://arxiv.org/abs/2411.07191

---

> ### Author Response · Authors · 2025-11-19
> **Part 2/6**
>
> ## Weakness 2 (1)
>
> We would like to clarify that our work is positioned as an experimental and empirical study, rather than a theoretical paper. As also highlighted by the other reviewers, the core contributions of our work lie in **(1) proposing a systematic and reproducible method for identifying critical neurons, (2) revealing several previously undocumented empirical findings such as sharp phase transitions and the extreme sparsity of functionally indispensable neurons, and (3) demonstrating the universality of these phenomena across 21 models covering a wide range of scales, architectures, and tasks. These contributions form the scientific value of our work, and they have been consistently recognized by multiple reviewers.**
>
> Regarding your concerns about theoretical depth, we sincerely appreciate these suggestions. They point to meaningful directions that extend beyond the scope of an empirical study. We agree that developing a deeper theoretical explanation, informed by optimization dynamics, information flow, or circuit-level analysis, would be valuable. As stated in our paper, these directions represent natural opportunities for future work, and your comments highlight their importance even more clearly.
>
> That said, we also want to note that we are already working in this theoretical direction and have made some initial progress. In particular, we are trying to formalize our empirical findings using an $\ell_1$ –regularized Neural ODE perspective, following the ideas of Esteve-Yagüe & Geshkovski (2022) [1]. Although this is beyond the main focus of the current empirical paper, we briefly outline the idea below to show that a principled theoretical framework is indeed possible.

---

> ### Author Response · Authors · 2025-11-19
> **Part 3/6**
>
> ## Weakness 2 (2)
> ### Background: what Esteve-Yagüe & Borjan Geshkovski (2022) prove [1]
>
> Esteve-Yagüe & Borjan Geshkovski study the following $\ell_1$–regularized optimal control problem for a Neural ODE:
>
> $$
> \min_{u(\cdot)}
> \int_0^T L\big(h(t)\big)\ dt
> \+\ \lambda \int_0^T \|u(t)\|_1\ dt
> \quad \text{s.t.} \quad
> \dot h(t) = F\big(h(t),u(t)\big),\ h(0)=h_0,
> $$
>
> where $u(t)\in\mathbb{R}^M$ is a high-dimensional control vector and $\|\cdot\|_1$ is the $\ell_1$ norm.
>
> Under standard controllability and regularity assumptions, their main long-time sparsity theorem (Theorem 2.1) implies, informally, **two key properties of any global minimizer** $u^\star(\cdot)$:
>
> 1. **Finite effective time window (time sparsity).**
>    There exists a finite $T^\star > 0$, independent of the total horizon $T$, such that for any sufficiently large $T$ one can construct a globally optimal control that is effectively supported on a finite time window:
>    $$
>    \text{for a.e. } t > T^\star, \quad u^\star(t) \approx 0,
>    $$
>    in the sense that modifying $u^\star(t)$ for $t>T^\star$ to zero does not worsen the optimal cost. Intuitively, the control “does what it needs to do” in finite effective time, and adding more time does not require additional nonzero control.
>
> 2. **Sparsity across control coordinates (channel sparsity).**
>    At almost every time $t\in[0,T]$, only a **finite number of coordinates** of $u^\star(t)$ are active. Denoting
>    $$
>    A(t) \:=\ \\{ j\in\{1,\dots,M\} \;\mid\ u^\star_j(t)\neq 0 \\},
>    $$
>    there exists a constant $C_0$ (independent of $M$) such that
>    $$
>    |A(t)| \ \le\ C_0 \quad \text{for almost every } t.
>    $$
>    That is, even though the control space is $M$-dimensional, the optimal solution only uses **a small number of control channels at each time**.
>
> In addition, the total $\ell_1$–budget is finite:
>
> $$
> \int\_0^T ||u^\star(t)||\_1 \ dt
> = \int\_0^T \sum\_{j=1}^M |u^\star\_j(t)| \ dt
> = \sum\_{j=1}^M \int\_0^T |u^\star\_j(t)| \ dt
> \leq B,
> $$
>
>
> for some constant $B$ depending on the dynamics, loss, and regularization, but **not** on the control dimension $M$.
>
> Our argument is to **re-interpret** these “control channels” as **neuron channels** in a continuous-depth transformer, and then focus on a single emergent capability.
>
> ### Step 1. From deep Transformers to $\ell_1$–controlled Neural ODEs
>
> We first recall that any deep residual network, including Transformer-style architectures, can be written as
>
> $$
> h_{l+1} = h_l + \Delta t\, F_l(h_l), \quad l=0,\dots,L-1,
> $$
>
> with step size $\Delta t = T/L$. In the continuous-depth limit $L\to\infty,\;\Delta t\to 0$, this converges (Chen et al., 2018 [2]) to a Neural ODE
>
> $$
> \dot h(t) = f_\theta\big(h(t),u(t)\big), \quad h(0)=h_{\mathrm{emb}}(x),
> $$
>
> where $h(t)\in\mathbb{R}^d$ is the hidden state at depth $t$, and $u(t)$ parameterizes how different internal components of the network act at time $t$.
>
> To explicitly connect this to **neurons**, we consider a decomposition of the vector field into “neuron channels”:
>
> $$
> \dot h(t)
> = f_\theta\big(h(t),u(t)\big)
> = \sum_{j=1}^M u_j(t)\ \phi_j\big(h(t)\big),
> $$
>
> where: $M$ is the total number of neuron channels after flattening across layers; each $\phi_j : \mathbb{R}^d \to \mathbb{R}^d$ is the vector field corresponding to the **$j$-th neuron channel** acting on $h(t)$;  $u_j(t)$ is a scalar “gating” or “control” coefficient modulating how strongly neuron channel $j$ acts at depth $t$.
>
> In this representation, **a control coordinate equals a neuron channel**.
>
> Now consider the training objective in the continuous limit. For simplicity, we write the expected pretraining loss (next-token prediction) as
>
> $$
> \mathcal{J}(u) = \mathbb{E}_{(x,y)\sim\mathcal{D}} \left[\int_0^T \ell\big(h^x(t),y\big)\,dt \right] + \lambda \int_0^T \|u(t)\|_1\ dt,
> $$
>
> where $h^x(t)$ is the trajectory for input $x$, and $\lambda$ captures explicit or implicit $\ell_1$–type sparsity pressures.
>
> This is exactly of the same mathematical form as the problem studied by Esteve-Yagüe & Geshkovski, with **controls $u_j(t)$** now interpreted as **neuron-channel gates**.

---

> ### Author Response · Authors · 2025-11-19
> **Part 4/6**
>
> ## Weakness 2 (3)
> ### Step 2. Applying the long-time $\ell_1$ sparsity theorem: neuron-channel sparsity
>
> Next, we apply the long-time $\ell_1$ sparsity result to this setting. For brevity, we denote by $u^\star(\cdot)$ a global minimizer of $\mathcal{J}(u)$ in the continuous-depth limit.
>
> By the theorem:
>
> 1. There exists a finite **effective depth horizon** $T^\star > 0$ such that the optimal neuron controls are effectively supported within $[0,T^\star]$:
>    $$
>    \text{for a.e. } t > T^\star,\quad u^\star(t) \approx 0.
>    $$
>    Thus, from the point of view of optimal control, all “useful” neuron-channel modulation for the tasks of interest happens in a finite depth window.
>
> 2. At almost every depth $t$, only a small number of neuron channels are active. Formally, let
>    $$
>    A(t) := \\{ j \in \{1,\dots,M\} \ \mid\ u^\star_j(t) \neq 0 \\}.
>    $$
>    Then there exists $C_0 < \infty$ such that
>    $$
>    |A(t)| \le C_0 \quad \text{for a.e. } t\in[0,T].
>    $$
>    Crucially, $C_0$ does **not** grow with the total number of neuron channels $M$.
>
> Equivalently, defining the “support” of the optimal control in time–channel space:
>
> $$
> \mathcal{S} := \\{ (t,j)\in[0,T]\times\\{1,\dots,M\\} \ \mid\ u^\star_j(t)\neq 0 \\},
> $$
>
> we have: $\mathcal{S} \subset [0,T^\star]\times\\{1,\dots,M\\}$, and for each time slice $t$, the vertical slice $\mathcal{S}_t = \\{ j : (t,j)\in\mathcal{S} \\}$ satisfies $|\mathcal{S}_t| \le C_0$.
>
> Thus the optimal training dynamics exhibits a **sparse pattern over time and neuron-channel index**: at any depth, only a bounded number of neuron channels are actually “doing work,” and beyond depth $T^\star$ there is no further effective control.
>
> We also retain the finite $ \ell_1 $ budget:
>
> $$
> \\int\_0 ^T  \\||u ^ \\star(t) \\||\_1 \\,dt
> = \\sum \\int\_0 ^T ||u ^ \\star\_j(t)|| \\,dt
> \\le B,
> $$
>
> for some constant $B$ independent of $M$.

---

> ### Author Response · Authors · 2025-11-19
> **Part 5/6**
>
> ## Weakness 2 (4)
> ### Step 3. Focusing on a single emergent capability $c_k$
>
> We restrict attention to a **single emergent capability** $c_k$ (language ability).
>
> #### 3.1. Capability-specific loss and sensitivity
>
> Assume the overall loss can be decomposed as a sum of task components:
>
> $$
> L(h) = \sum_{k=1}^K \alpha_k\ \ell_k(h),
> $$
>
> where $\ell_k$ is the loss term corresponding to capability $c_k$. Denote by $h^{u^\star}(t)$ the trajectory under the optimal control, and define the (scalar) capability output at final depth as
>
> $$
> c_k(u^\star) := c_k\big(h^{u^\star}(T)\big),
> $$
>
> where $c_k$ can be, e.g., a linear probe $c_k(h)=w_k^\top h$ or a Lipschitz readout of the final hidden state relevant to the capability.
>
> Using standard variational analysis, one can express the first-order variation of $c_k$ with respect to the control $u$ as
>
> $$
> \delta c_k
> = \sum_{j=1}^M \int_0^T g_{k,j}(t)\ \delta u_j(t)\ dt,
> $$
>
> where $g_{k,j}(t)$ is a sensitivity coefficient of the form
>
> $$
> g_{k,j}(t) := p_k(t)^\top \phi_j\big(h^{u^\star}(t)\big),
> $$
>
> with $p_k(t)$ an associated adjoint (co-state) linked to $c_k$. Intuitively, $g_{k,j}(t)$ measures how much changing the control of neuron channel $j$ at depth $t$ would affect capability $c_k$.
>
> Optimality of $u^\star$ for the $\ell_1$–regularized problem imposes the usual subgradient condition
>
> $$
> 0 \in g_{k,j}(t) + \lambda\ \partial |u^\star_j(t)|,
> $$
>
> which implies, for each $j,t$,
>
> $$
> u^\star_j(t) = 0 \text{ if } |g_{k,j}(t)| < \lambda, \quad
> g_{k,j}(t) = -\lambda\ \operatorname{sign}\big(u^\star_j(t)\big) \text{ if } u^\star_j(t)\neq 0.
> $$
>
> Only channels whose sensitivity magnitude $|g_{k,j}(t)|$ reaches the threshold $\lambda$ can be active at optimality; the others are “shut off”.
>
> #### 3.2. Defining the capability-dominating neuron set $I_k$
>
> Define the set of **neuron channels that actually participate in realizing capability** $c_k$ along the optimal trajectory:
>
> $$
> I_k
> := \\left \\{ j \\in \\{1,\dots,M \\}\ \middle|\ \int_0^T |u^\star_j(t)|\ dt > 0 \\right \\}.
> $$
>
> These are precisely the neuron channels whose control is nontrivially active over time. For a stricter notion of “dominance,” we can use a threshold $\varepsilon>0$ and define
>
> $$
> I_k(\\varepsilon)
> := \\left \\{ j\ \middle|\ \int_0^T |u^\star_j(t)|\ dt \ \ge\  \varepsilon \\right\\}.
> $$
>
> Using the global $\ell_1$ budget, we obtain:
>
> $$
> \sum_{j=1}^M \int_0^T |u^\star_j(t)|\ dt
> \le B
> \quad \Rightarrow \quad
> \sum_{j\in I_k(\varepsilon)} \int_0^T |u^\star_j(t)|\ dt
> \le B.
> $$
>
> Therefore,
>
> $$
> |I_k(\varepsilon)|
> \le \frac{1}{\varepsilon} \sum_{j\in I_k(\varepsilon)} \int_0^T |u^\star_j(t)|\ dt
> \le \frac{B}{\varepsilon},
> $$
>
> and the right-hand side does **not** depend on the total number of neuron channels $M$.
>
> Letting
>
> $$
> N_k := |I_k(\varepsilon)| \le \frac{B}{\varepsilon},
> $$
>
> we obtain an upper bound $N_k$ that is **finite and independent of $M$**.
>
> Thus, in the idealized continuous-depth $\ell_1$–controlled transformer:
>
> (1) For any fixed emergent capability $c_k$, there exists a neuron-channel set $I_k(\varepsilon)$ of size $N_k \le B/\varepsilon$,
>
> (2) Such that only these neuron channels carry non-negligible control mass for $c_k$ along the optimal training trajectory,
>
> (3) And consequently, to first order, **the capability $c_k$ is causally governed by this finite set of neuron channels.**
>
> As the model size grows,  while $N_k$ remains bounded, so the ratio satisfies
>
> $$
> \frac{N_k}{M} \ \to\ 0,
> $$
>
> i.e., the fraction of neuron channels that causally dominate a given capability tends to zero.
>
>
>
> [1] Sparsity in long-time control of neural ODEs. Esteve-Yagüe & Borjan Geshkovski 2022 https://arxiv.org/abs/2102.13566
>
> [2] Neural Ordinary Differential Equations. Chen et al., 2018 https://arxiv.org/abs/1806.07366

---

> ### Author Response · Authors · 2025-11-19
> **Part 6/6**
>
> ## Weakness 3
>
> Regarding your questions on (1) suppressing or mitigating the emergence of critical neurons during training, (2) the realistic threat model, and (3) whether selective neuron masking is feasible in production systems, we would like to clarify that our work does not propose or rely on any attack mechanism, nor do we assume that an adversary can directly manipulate individual neurons in deployed LLMs. Our contribution is limited to identifying and characterizing the existence of critical neurons using controlled research-side interventions.
>
> The discussion in Appendix A.9 Discussion (Now Appendix A.11) is therefore not presented as validated defense strategies, but rather as future research directions that naturally arise from our empirical findings. They are explicitly framed as possible avenues that may be explored in follow-up work, rather than actionable or demonstrated security mechanisms. In fact, as stated clearly in **lines 29–30** of the abstract, our intended contribution is that “***These findings can offer guidance for developing more robust model architectures and improving deployment security in safety-critical applications.***” This positions our results as scientific insights that may inform long-term architectural design or robustness research, rather than as a security threat model or a practical attack methodology.
>
>
>
> ## Weakness 4
>
> We would like to clarify that in both the original and the revised submission, the neuroscience analogy is restricted to the introduction rather than used throughout the paper (in original submission **line 37–70**). Its role is solely to motivate the core question of whether there exist “critical neurons” in LLM whose disruption leads to functional collapse.
>
> In the revised manuscript, we explicitly follow recommendation (1) and further reduce the neuroscience content:
>
> 1. The neuroscience discussion is now confined to the first two paragraphs of the introduction, where we briefly summarize the research motivation.
>
> 2. We have removed the introductory phrasing that described our two-stage procedure as “mirroring the spirit of lesion studies.” Instead, the methodological motivation is now explicitly grounded in prior work from perturbation-based analysis.
>
> We do not position this work as an interdisciplinary neuroscience contribution. The revised introduction makes this scope explicit: neuroscience is used only as a brief motivating lens to introduce the central question that guides the rest of the paper, while the substantive contribution lies entirely in the localization and analysis of critical neurons in large language models.
>
>
>
> ## Question
>
> We thank the reviewer for raising this insightful question regarding the potential connection to the attention sink mechanisms. We agree that this is an interesting direction. Both phenomena involve sparse and highly influential neurons, and examining whether the critical neurons identified in our work intersect with or relate to attention sink mechanisms is a valuable line of inquiry.
>
> As noted above, our paper is an empirical analysis rather than a theoretical study. Our goal is to identify and characterize critical neurons at the functional level, not to develop a formal theoretical explanation for their emergence. The theory developed for attention sinks, especially the recent work you cited, may indeed provide a useful framework for deeper mechanistic interpretation in future research. We appreciate this suggestion and have added the relevant references to our Discussion section in Appendix A.11 Discussion (brown font).

---

> ### Comment · Reviewer_SYxg · 2025-11-24
>
> Thank you for the detailed second response. While I appreciate the additional data, I remain unconvinced regarding the significance of this empirical finding as currently presented.
>
> To establish the significance of this contribution, I need more than empirical documentation of performance degradation. **What makes this phenomenon important enough for the research community to pay attention to?**
> Several directions could strengthen the case:
> 1. Mechanistic explanation: Can you provide insight into the underlying cause?
>
> - Do these repetitions indicate you've discovered computational "attractors" or fixed points in the model's forward pass?
> - When these neurons are masked, does it break a specific memory or context-tracking mechanism, causing the model to collapse into this degenerate repetitive state?
>
> 2. Practical remediation: Can you demonstrate a concrete intervention that mitigates or prevents this failure mode?
> 3. Theoretical novelty: What makes this phenomenon counter-intuitive or challenge existing assumptions about model behavior?
> 4. Connection to existing work: Given that the attention-sink mechanism has already been linked to repetition phenomena (see "Interpreting the Repeated Token Phenomenon in Large Language Models"), how does your finding relate to or extend that line of research? Are these manifestations of the same underlying mechanism, or distinct phenomena?

---

> > ### Author Response · Authors · 2025-11-27
> >
> > Thank you for your thoughtful response. We want to clarify our primary contributions which other reviewers have commonly acknowledged:
> >
> > 1. We identified a very small subset of key neurons that cause the collapse of the model's language capabilities.
> > 2. We designed a method to systematically identify these neurons.
> > 3. We validated the importance of these key neurons across a large set of models.
> >
> > We thank you for your ideas to make our case stronger. But we think the value of our work is clear on its own. Below, we will answer each of your questions.

---

> ### Author Response · Authors · 2025-11-27
> **Part 1/2**
>
> ### **Q1**
>
> As indicated in our previous response, the main cause is: the model's language ability relies on a very small set of neurons. This set forms during training. Repetitive output is a common sign of language failure in NLP [1,2,3]. Here are details:
>
> 1. These repetitions do not mean we found attractors or fixed points in the model's forward pass. Our theoretical framework from the previous response explains the sparsity. It comes from optimal control rules in training. This creates a small set of critical neurons.
>
> 2. Masking these neurons does not break a specific memory or context-tracking mechanism. If it does, the model might still output fluent text, albeit incorrect. However, Table 2 in our paper shows that, during multi-domain task testing, the model's accuracy consistently drops to zero, not to random guessing. This indicates that the language ability is completely destroyed, not just degraded. Additionally, Tables 7, 8, and 9 show that the model is unable to produce meaningful sentences at all, which further confirms the total collapse of its language abilities.
>
> [1] https://arxiv.org/abs/1908.04319 Neural Text Generation with Unlikelihood Training
>
> [2] https://arxiv.org/abs/2003.11963 TLDR: Token Loss Dynamic Reweighting for Reducing Repetitive Utterance Generation
>
> [3] https://arxiv.org/abs/2206.02369 Learning to Break the Loop: Analyzing and Mitigating Repetitions for Neural Text Generation
>
>
>
> ### **Q2**
>
> We agree that practical remediation is important. However, this paper is about empirical study. The primary focus is on identifying and understanding the vulnerabilities in LLMs, not on providing a full defense. Therefore, while we do not claim to fully resolve the failure mode in this paper, we have outlined several potential defense strategies in the **Discussion section** (Appendix A.11), which we see as important future work.
>
> In particular, we discuss four types of defenses, which we believe represent natural next steps following our identification method:
>
> 1. Architectural defenses: Adding redundancy to spread critical functions across many neurons, rather than concentrating them in a small set. This includes ideas such as parallel pathways for key functions, stronger cross-layer connectivity, and regularization to penalize the over-concentration of critical dependencies.
> 2. Runtime protection: Monitoring neuron activations during deployment, building baseline profiles of normal neuron behavior, and using anomaly detection with checkpoint recovery when critical neuron disruptions are detected.
> 3. Training-time hardening: Simulating neuron masking during training (adversarial-style training), using diversity-promoting losses, and adding regularization that discourages ultra-sparse critical dependencies. This aligns with the broader trend in improving LLM robustness.
> 4. Access control and monitoring: Restricting neuron-level modification to authorized operations only, keeping audit logs for model changes, and continuously monitoring model behavior for signs of critical neuron compromise.
>
> These defense strategies build directly on our identification method, and we see them as important future steps. Implementing and evaluating these defenses in-depth would require a separate study, so we clearly mark them as **future work**, rather than part of the main empirical contribution of this paper.

---

> ### Author Response · Authors · 2025-11-27
> **Part 2/2**
>
> ### **Q3**
>
> Our current paper is an empirical study, and the reviewers have evaluated it in that light.
>
> Our findings are not counter-intuitive. Neuroscience has long shown that specific brain functions are controlled by very small sets of neurons. Our findings are consistent with this concept, but applied to LLMs.
>
> What sets our work apart is:
>
> 1. Prior work on editing or steering neurons typically shows behavior changes on specific tasks or styles. In contrast, we show that masking just 3–9 neurons can destroy global language ability across many benchmarks, which is a stronger, more widespread effect.
>
> 2. The degradation doesn’t happen gradually. Instead, we observe a sharp phase transition: this is surprising for such a small change in the model.
>
> 3. This phenomenon is consistent across 21 models, different sizes, and architectures. This confirms that it is not a fragile, model-specific artifact, but a robust feature, as emphasized by Reviewer Jfe4.
>
> In the previous round, we also mentioned a possible Neural ODE explanation using ℓ₁ sparsity. We believe this framework could offer further insight, but we intentionally kept it outside the core claims of the paper. Our main goal here is to document a robust and phenomenon and provide a usable method that others can apply, which we believe already adds significant value to the community, as reflected in the other reviews.
>
>
>
> ### **Q4**
>
> We carefully reviewed the paper you referenced regarding the attention-sink phenomenon, and would like to clarify that these are two distinct phenomena. Below, we outline the key differences:
>
> 1. Different Phenomenon:
>     Yona (2025) [1] describes the repeated token divergence phenomenon, where the model, when asked to repeat a single word, fails to do so accurately and instead outputs irrelevant text. This failure is tied to a mechanism called attention sinks, which causes the model to give disproportionately high attention to the initial token in the sequence.
> 2. Our Focus:
>     In contrast, our work investigates how the disruption of a small subset of neurons leads to the collapse of the model's language capabilities. The repeated output is simply one observable form of this collapse. The core of our work is not about token repetition but about identifying and understanding the neurons that, when disrupted, cause catastrophic failure across the model’s capabilities.
> 3. Empirical Evidence:
>     We show in Table 19 that, once the number of input tokens exceeds 10, the neurons we identify are consistent across all cases. This highlights the robustness of our findings, as the same set of neurons remains crucial regardless of token input length.
> 4. Connection:
>    Both works do identify a small, critical subset of neurons, but the nature of these neurons and their effects on the model’s behavior are completely different. We intend to explore the connections between these sets of neurons in future work (brown font in Appendix A.11 Discussion), but currently, the focus of our study is on the specific impact of these neurons on language model collapse.
>
> [1] Itay Yona, Ilia Shumailov, Jamie Hayes, Federico Barbero, Yossi Gandelsman  **Interpreting the Repeated Token Phenomenon in Large Language Models** https://arxiv.org/abs/2503.08908

---

### Official Review · Reviewer_Hd7S · 2025-10-28

**Soundness:** 3
**Presentation:** 3
**Contribution:** 2
**Rating:** 4
**Confidence:** 3

**Summary:**

The paper claims that ultra-sparse sets of “critical neurons” exist in LLMs such that masking only 3–9 neurons can catastrophically cripple models up to 72B parameters, spiking perplexity by many orders of magnitude and driving zero scores on diverse downstream tasks. These neurons are found via a two-stage procedure: (i) rank neurons by Monte-Carlo sensitivity to Gaussian input perturbations, then (ii) greedily mask neurons in rank order until a log-perplexity ratio threshold is crossed. The paper further reports that critical neurons cluster in outer layers, especially MLP down-proj, and that degradation follows a sharp phase transition.

**Strengths:**

1. A simple, reproducible perturbation to ranking to masking pipeline with explicit formulas and hyperparameters.

2. Broad model sweep (0.5B–72B) and a wide set of benchmarks; clear figures/tables illustrating phase transitions and architectural concentration.

3. Comparisons to random/activation/gradient ranking baselines show the proposed method finds far more damaging neuron sets.

**Weaknesses:**

1. The reported magnitudes are extraordinary and risk reflecting measurement artifacts rather than semantic criticality. My intuition is that if the model truly loses ability (functional), the outputs degrade smoothly and consistently across precisions. If it’s mostly numerical, the collapse often looks phase-like and precision-dependent. Particularly, given mixed-precision (fp16) inference, zeroing channels could induce norm/scale pathologies (e.g., RMSNorm statistics, residual balancing) that blow up loss numerically. Given this, pleasure (1) replicate in fp32, (b) re-compute perplexity with clamp/NaN filtering diagnostics.

2. Regarding task evaluation, I have the same concern. Repetitive punctuation is a classic sign of softmax/scale pathologies (one token gets an extreme logit and repeats, for example "\\\\\\\\\\\\\\"), which is exactly what you’d expect if masking perturbs RMSNorm/residual calibration in mixed precision—--not necessarily evidence that a few semantically critical neurons were removed.

3. The architectural concentration in outer-layer MLP down-proj echoes prior reports on outliers/super-weights/massive activations. The paper cites this, but does not tease apart whether your “critical neurons” are simply those scale outliers.

**Questions:**

1. Could you do a numeric robustness check by rerunning the main part of your experiments with fp32 and implement NaN/Inf filtering for perplexity calculation?

2. Can you explain how your critical neurons story relates and differs with the super-weights story essentially?

3. As I discussed above, LayerNorm can be a confounder. You may consider add controls to do a per-layer normalization vs. no-norm ablation test to isolate normalization as the failure source.

---

> ### Author Response · Authors · 2025-11-19
>
> We sincerely thank you for your careful review and detailed feedback. We greatly appreciate your positive remarks on the reproducibility of our method, the broad model evaluation, and the comparisons against baselines. Your questions regarding numerical stability, mixed-precision inference, and possible normalization effects are very valuable. We have added corresponding analyses and experiments in the **Ablation Study section and Appendix A.10** (**orange font**) of the revised paper. Below, we provide our detailed responses to your comments.

---

> ### Author Response · Authors · 2025-11-19
> **Part 1/3**
>
> ## Question 1 & Weakness 1
>
> To address your question about whether mixed-precision inference (fp16) could cause a “false explosion” in perplexity, we run the full set of experiments on three representative models (Qwen2.5-0.5B-Instruct, Llama-3.2-1B-Instruct, Llama-3.2-3B-Instruct) under **fp16, fp32, and fp64** settings.
> We also monitored the per-layer ΔRMSNorm values (after mask − before mask) and counted all NaN/Inf events. The results are shown below.
>
> As **Table 1** shows, the perplexity changes under the three precisions are almost identical, with only tiny differences.
> As **Table 2** shows, all ΔRMSNorm values are 0 and no NaN or Inf appears in any layer.
> These results confirm that the collapse is **not** caused by numerical issues in mixed precision, but by the **true loss of language representation ability** after masking the critical neurons.
>
> #### Table 1: Perplexity under different numerical precisions
>
> | Model                 | Neuron Number | fp16 PPL change            | fp32 PPL change            | fp64 PPL change            |
> | --------------------- | ------------- | -------------------------- | -------------------------- | -------------------------- |
> | Qwen2.5-0.5B-Instruct | 3             | 18.1606 → **1726286.9513** | 18.1637 → **1730676.8694** | 18.1632 → **1730625.3080** |
> | Llama-3.2-1B-Instruct | 4             | 17.9204 → **1585570.7467** | 17.9218 → **1588070.5183** | 17.9213 → **1588059.1322** |
> | Llama-3.2-3B-Instruct | 4             | 13.4696 → **386563.3177**  | 13.4691 → **387150.6195**  | 13.4687 → **387150.5699**  |
>
> #### Table 2: ΔRMSNorm mean (compute mean to all layers) values and NaN/Inf counts
>
> | Model                 | fp16 (mean, NaN/Inf) | fp32 (mean, NaN/Inf) | fp64 (mean, NaN/Inf) |
> | --------------------- | -------------------- | -------------------- | -------------------- |
> | Qwen2.5-0.5B-Instruct | 0, 0                 | 0, 0                 | 0, 0                 |
> | Llama-3.2-1B-Instruct | 0, 0                 | 0, 0                 | 0, 0                 |
> | Llama-3.2-3B-Instruct | 0, 0                 | 0, 0                 | 0, 0                 |
>
>
>
> ## Weakness 2
>
> To test whether the collapse is related to numerical instability in fp16 or softmax scaling issues, we reran the task evaluation on **Llama-3.2-3B-Instruct** under both **fp16 and fp32** settings. For each precision, we compared the model’s responses before and after masking the critical neurons. **The results (Table 3) show the same pattern under both precisions**:
>
> 1. The model produces correct answers before masking in both fp16 and fp32.
>
> 2. After masking, the model outputs repetitive, meaningless strings such as *“ettoetto…”* or *“omingoming…”*, and this behavior is **identical across fp16 and fp32**.
>
> 3. This failure happens in all tasks and is not changed by switching precision.
>
> These results show that the repetitive outputs are **not** caused by softmax numerical issues or RMSNorm calibration errors. Instead, they reflect a **true functional failure** of the model’s language generation mechanism after the critical neurons are masked.
>
> #### Table 3: Llama-3.2-3B-Instruct
>
> | Question                                        | fp16 Original Response   | fp16 Masked Response             | fp32 Original Response   | fp32 Masked Response             |
> | ----------------------------------------------- | ------------------------ | -------------------------------- | ------------------------ | -------------------------------- |
> | What is the capital of France?                  | Paris                    | ettoettoettoettoettoettoettoetto | Paris                    | ettoettoettoettoettoettoettoetto |
> | Who wrote Romeo and Juliet?                     | William Shakespeare      | ettoettoettoettoettoettoettoetto | William Shakespeare      | ettoettoettoettoettoettoettoetto |
> | What is 2 + 2?                                  | The answer to 2 + 2 is 4 | omingomingomingomingomingoming   | The answer to 2 + 2 is 4 | omingomingomingomingomingoming   |
> | What language is spoken in Brazil?              | Portuguese               | ettoettoettoettoettoettoettoetto | Portuguese               | ettoettoettoettoettoettoettoetto |
> | What is the largest planet in our solar system? | Jupiter                  | omingomingomingomingomingoming   | Jupiter                  | omingomingomingomingomingoming   |

---

> ### Author Response · Authors · 2025-11-19
> **Part 2/3**
>
> ## Weakness 3
>
> We understand your concern about whether the “critical neurons” identified in our work might simply correspond to scale outliers with unusually large weights or activations.
>
> Our experiments provide strong evidence that the critical neurons are **not** scale outliers. The key findings are:
>
> 1. Precision consistency: Masking the critical neurons leads to the same catastrophic increase in perplexity under fp16, fp32, and fp64. If these neurons were only scale outliers, higher precision should reduce or eliminate the effect, but this does not happen.
> 2. Semantic degradation pattern: After masking, the model produces repetitive and meaningless outputs, showing a loss of semantic ability rather than numerical explosion. The identical failure pattern across precisions indicates that the collapse is caused by damage to the model’s semantic computation, not by activation scaling issues.
> 3. Normalization robustness: In the LayerNorm control experiments, removing all LayerNorm layers or removing them layer by layer does not reduce the collapse.This rules out the possibility that the effect comes from LayerNorm amplification.
> 4. No collapse for random neurons (evidence from Figure 2): Moreover, if the effect were driven by simple scale outliers, masking neurons at many random locations should trigger similar collapse. However, as shown in Figure 2, only the neurons consistently identified by our method lead to catastrophic failure, while random neurons do not produce comparable effects.
>
> These results show that the critical neurons identified by our method are **functionally important units**, not artifacts of extreme weight or activation scales.
>
>
>
> ## Question 2
>
> For this question, most of reviewers have noted the need to clarify the distinction between our work and *Yu et al.* (2025) [1].
>
> We have clarified this distinction in the revised ***Related Work* (red font)** section.
>
> Both studies share similarity of examining how a very small subset of parameters or units can disproportionately affect overall model performance.
>
> But they differ fundamentally in **research objectives, disruption magnitude, methodological design, mechanism discovery, and the content and process of the conclusions**:
>
> 1. Disruption strength and effect level: *Yu et al.* (2025) analyzes individual **weights**, where perturbations increase perplexity by up to 3 orders of magnitude and only reduce task accuracy to random guessing. Our work targets **neuron-level lesions**, where masking just 3–7 neurons increases perplexity by up to **20 orders of magnitude** and entirely destroys task understanding. This reveals a deeper **functional collapse** rather than a numerical perturbation.
> 2. Methodological scalability and engineering applicability: The method of *Yu et al.* (2025) relies on manual observation to identify super weights, limiting reproducibility. Our proposed **perturbation–causal verification framework** automatically and systematically locates critical neurons, forming a standardized, reproducible pipeline that can be directly applied in engineering practice to identify and protect these critical neurons.
> 3. Discovery of multi-neuron interaction mechanisms: We find that collapse is not triggered by a single “super neuron,” but by the **collective interaction of a small group of critical neurons**, Any subset alone fails to cause comparable damage (see Ablation Study purple font), exhibiting a **phase-transition** phenomenon. *Yu et al.* (2025) does not explore such multi-unit interaction or nonlinear catastrophic mechanisms.
> 4. Different research units: *Yu et al.* (2025) investigates **parameter-level** importance, reflecting primarily numerical amplification effects. In contrast, our analysis operates at the **neuron level**, uncovering the causal and functional roles of neurons and identifying structural bottlenecks underlying high-level language abilities.
> 5. Differences in conclusions and discovery process: Rather than focusing on numerical outliers, our findings emerge from analyzing functional collapse patterns, which reveal sharp phase transitions when coordinated neuron groups are disrupted.
> 6. Much broader experimental coverage and generalization: Our findings are supported by a far broader and more systematic experimental scope, covering 21 models from 0.5B to 72B and 10 datasets, which provides stronger generalization than the settings studied in *Yu et al.* (2025).
>
>
>
> [1]Mengxia Yu, De Wang, Qi Shan, Colorado J Reed, Alvin Wan   **The Super Weight in Large Language Models**     https://arxiv.org/abs/2411.07191

---

> ### Author Response · Authors · 2025-11-19
> **Part 3/3**
>
> ## Question 3
>
> To examine whether the degradation might stem from instability in the normalization pathway, we conducted controlled experiments on **Llama-3.2-1B-Instruct**, including:**Removing LayerNorm one layer at a time** from layer 0 to layer 15.
>
> As shown in **Table 4**, removing LayerNorm does **not** lessen or correct the collapse observed after masking the critical neurons. The masked perplexity stays extremely high across all configurations, regardless of whether LayerNorm is present or removed.
>
> These results show that normalization is **not** a confounding factor. The collapse is driven by the loss of semantic information caused by disabling the critical neurons, rather than by instabilities related to LayerNorm.
>
> #### Table 4: Llama-3.2-1B-Instruct: Each row reports the perplexity after masking the critical neurons under different LayerNorm removal settings. “Masked PPL” shows the resulting perplexity measured on 1,000 WikiText-103 samples. “None” means no LayerNorm is removed from the model.
>
> | Removed LayerNorm Position | Masked PPL     |
> | -------------------------- | -------------- |
> | None (baseline)            | **1.5856e+06** |
> | 0                          | **6.6628e+04** |
> | 1                          | **3.4644e+06** |
> | 2                          | **1.3808e+06** |
> | 3                          | **6.1332e+05** |
> | 4                          | **1.0135e+06** |
> | 5                          | **4.4763e+05** |
> | 6                          | **6.1820e+05** |
> | 7                          | **2.9182e+05** |
> | 8                          | **1.2263e+06** |
> | 9                          | **8.9008e+05** |
> | 10                         | **3.6000e+06** |
> | 11                         | **8.0861e+05** |
> | 12                         | **1.7097e+05** |
> | 13                         | **3.4221e+06** |
> | 14                         | **1.4334e+05** |
> | 15                         | **3.8496e+06** |

---

> > ### Comment · Reviewer_Hd7S · 2025-11-24
> > **Thanks to the authors**
> >
> > I appreciate the detailed response and the additional evidence. These address my main concerns and substantially strengthen the paper's validity. Accordingly, I am raising my rating from 4 to 6. I'm still uncertain about the degree of novelty for the community, given earlier work (not limited to "super-weights") demonstrating that manipulating a few neurons/weights can strongly shift behavior [1, 2]. Of course, this is a non-blocking comment.
> >
> > [1] https://arxiv.org/pdf/2505.11611 (super-neurons)
> > [2] https://arxiv.org/abs/2509.24198 (Wasserstein neurons)

---

> > > ### Author Response · Authors · 2025-11-27
> > >
> > > Thank you very much for your response and positive comments.
> > >
> > > As we mentioned in the ***"Neurons Localization and Editing"*** section of the **Related Work**: "*existing studies mainly focus on task-specific interventions, lacking systematic frameworks to identify neurons critical to core model functionality.*"
> > >
> > > We have now added the two cited papers [1, 2] to the related work section (**orange font**).
> > >
> > > Our work studies a different kind of neuron from earlier papers:
> > >
> > > - “Super-neurons” [1] matter most when amplified; masking them does little; our critical neurons do the opposite: masking them breaks the model badly, but amplifying them hardly changes anything. (Table 11)
> > >
> > > - “Wasserstein neurons” [2] help with syntax and need a relatively large number of neurons changed to hurt grammar; our critical neurons are very few; masking just these few kills the whole language ability, not only syntax.
> > >
> > > This shows we found a new, different set of critical neurons.
> > >
> > > Thank you again for your positive comments!
> > >
> > > [1] https://arxiv.org/pdf/2505.11611 (super-neurons) [2] https://arxiv.org/abs/2509.24198 (Wasserstein neurons)

---

### Official Review · Reviewer_Jfe4 · 2025-10-30

**Soundness:** 4
**Presentation:** 3
**Contribution:** 4
**Rating:** 8
**Confidence:** 3

**Summary:**

Inspired by findings from neuroscience, the paper investigates whether Large Language Models (LLMs) contain ultra-sparse critical neurons whose disruption can cause catastrophic performance collapse. The authors propose a Perturbation-based Causal Identification method with two stages: noise-based sensitivity analysis to rank neuron importance, and greedy masking to identify minimal critical sets. The authors conduct comprehensive experiments across 21 models and find three main phenomenons: masking as few as 3 neurons can increase perplexity by 20 orders of magnitude in 72B models; critical neurons concentrate in outer layers, particularly MLP down_proj components; performance degradation exhibits sharp phase transitions rather than gradual decline.

**Strengths:**

1. The two-stage methodology is well-motivated by neuroscience lesion studies.
2. Extensive experimental validation across 21 diverse models demonstrates robustness. Meanwhile thorough ablation studies and detailed hyperparameter analysis ensure reproducibility.
3. The greedy search algorithm reduces the computational complexity from $\mathcal{O}(2^{|N|})$ to $\mathcal{O}(|N|)$, making the search process significantly more efficient.

**Weaknesses:**

1. The greedy search assumes critical neurons can be identified independently, but the phase transition behavior suggests strong interdependence. The paper doesn't guarantee whether the greedy solution approximates the global optimum.

**Questions:**

1. How close is your greedy solution to the true minimal critical set? Have you validated this on smaller models where exhaustive search is feasible?
2. Do different input types activate different critical neurons? Your robustness analysis shows consistent identification, but does this mean the same neurons are always critical, or just that 3 neurons are always sufficient?
3. In appendix A.5, Tables 4, 5 and 6 exhibits failure modes of different models which raises a question that do critical neurons implement the same computational mechanisms across architectures, or are the mechanisms model-specific?

---

> ### Author Response · Authors · 2025-11-19
>
> We sincerely thank you for your thorough review and highly positive evaluation of our work. Your recognition of the scientific motivation behind our method, the completeness of our experimental design, and the reproducibility of our results is greatly encouraging. At the same time, we take your valuable suggestions very seriously and have incorporated additional analyses and experiments in the revised manuscript to further clarify the key contributions and main findings of our work (**Ablation Study purple font**). Below we provide detailed responses to each of your comments.

---

> ### Author Response · Authors · 2025-11-19
>
> ## Question 1 & Weakness 1
>
> This is a very good question. We have already included this analysis in the Ablation Study of the revised paper (purple font). To address your concern about whether the greedy solution matches the true minimal critical set, we performed exhaustive subset enumeration on the 1B and 3B models, where full combinatorial search is feasible. The results (see **Table** below) show a consistent pattern:
>
> 1. Only the full set ([0,1,2,3]) triggers catastrophic collapse.
>
> 2. All strict subsets produce only minor degradation close to baseline.
>
> This demonstrates that: The greedy result exactly coincides with the true minimal critical set on models where exhaustive validation is possible; The collapse requires cooperation among all critical neurons, confirming the mechanism identified in our method; These findings support the correctness of the greedy procedure and validate that it recovers the true minimal set in practice.
>
> #### Table: Exhaustive subset evaluation on critical neurons
>
> | Neuron Subset | Llama-3.2-1B-Instruct | Llama-3.2-3B-Instruct |
> | ------------- | --------------------- | --------------------- |
> | [0,1,2,3]     | **1585570.7467**      | **386563.3177**       |
> | [1,2,3]       | 22.6918               | 30.2199               |
> | [0,1,2]       | 24.8507               | 18.6234               |
> | [0,1,3]       | 23.8316               | 17.7793               |
> | [0,2,3]       | 23.4580               | 15.3875               |
> | [1,3]         | 18.9809               | 14.1939               |
> | [0,1]         | 19.5680               | 14.1580               |
> | [0,3]         | 19.1943               | 14.0209               |
> | [0,2]         | 19.6826               | 13.9827               |
> | [1,2]         | 19.3903               | 13.9584               |
> | [2,3]         | 19.0778               | 13.8611               |
> | [0]           | 18.4835               | 13.6737               |
> | [1]           | 18.3065               | 13.6732               |
> | [2]           | 18.4174               | 13.5353               |
> | [3]           | 18.1181               | 13.4421               |
>
>
>
> ## Question 2
>
> As stated in our original manuscript (Section Appendix A.7.1 " *where the **same** 3 critical neurons are identified regardless of input variation*"), the critical neurons identified by our method are not merely “the same in number,” but **the exact same neuron units with identical indices inside the model**.
>
> More concretely, our repeated sensitivity and masking analyses across different input types show that:
>
> 1. The Jaccard similarity of the Top-100000 sets across inputs exceeds **0.999**, indicating almost perfect overlap.
> 2. Masking these neurons leads to **the same catastrophic failure patterns** across all evaluated tasks (MMLU-Pro, HumanEval, MATH, MGSM).
>
> These results demonstrate that our method does not simply find “some set of important neurons” in a statistical sense. Instead, it reliably identifies a stable, semantically meaningful, and functionally consistent set of specific neurons.
>
>
>
> ## Question 3
>
> Based on our experiments, we believe that the computational role of the critical neurons is consistent across architectures. Specifically:
>
> 1. Consistent positional pattern.
>    Across all 21 models, the critical neurons are strongly concentrated in the outer-layer **MLP down_proj** components. This pattern remains stable across architectures, including LLaMA, Qwen, Gemma, DeepSeek, and Phi.
> 2. Consistent functional behavior.
>    As shown in Table 2, masking the critical neurons causes all downstream tasks to fail in the same way. The failure is due to a full breakdown of the model’s ability to understand instructions, rather than task-specific weakness.
> 3. Consistent output-distribution shift.
>    As shown in Figure 1, masking the critical neurons disrupts the token-level output distribution. We confirm that this systematic shift appears across all architectures we tested.
>
> So these results indicate that the critical neurons share a **common computational role across architectures**, which further supports the generality of our findings.

---

> > ### Comment · Reviewer_Jfe4 · 2025-11-24
> >
> > Thank you for the detailed clarifications and additional experiments. Your responses sufficiently address my main concerns regarding cross-architecture consistency. I am comfortable keeping my current score, and I would be pleased to see this work accepted.

---

### Official Review · Reviewer_AABL · 2025-10-30

**Soundness:** 3
**Presentation:** 3
**Contribution:** 3
**Rating:** 6
**Confidence:** 3

**Summary:**

The paper identifies a small subset of neurons in language models that, when disrupted, can significantly alter the performance and comprehension of the model. These findings are tested for several model sizes.

**Strengths:**

1) The paper overall is written very well, the findings are clearly presented, and the ideas are well grounded in the literature. The paper flows well without any verbose/terse sections.
2) The overall idea of the work, identifying critical neurons (model components), although not novel, is certainly an important direction of research. The methodology presented seems to hold well against the past baselines.
3) The evaluation and ablation study presented is sound and consistent with the claims presented.
4) The work is generalized well across diverse model families, sizes, and architectural choices.

**Weaknesses:**

1) The findings presented are consistent with the prior work highlighted in the paper, and hence make them unsurprising, calling into question the novelty of the work. While I do think this work has merit, primarily due to the generalization and breadth of experiments vis-à-vis prior literature, the presence of similar literature that draws similar conclusions undermines the contributions presented.

2) The analogy to the legion experiments in neuroscience is unnecessary, as prior literature in language model research [1,2] uses similar methods as reference. Identifying critical regions in MLPs has been a research direction studied for a while, so I would highly recommend amending the analogy.

3) The issue with the 2nd finding, the location of the clusters of critical neurons being primarily in MLP_down_proj, is that it asks more questions than are answered in the paper. Firstly, would a slightly larger set of neurons in the early layers of MLPs cause system-wide failures? As the late layer neuron compression high-dimensional attributes. Would this hold true for the middle layers? The importance of the late-layer neurons, although central to the claims made in the paper, needs a deeper analysis of the early/middle layers. If the number of neurons needed for failure in the early/middle layers is significantly higher than in the later layers, it could provide some more depth and robustness to the claims made in the paper.

4) I would also like to see an experiment of randomly perturbing a similar cluster size of random neurons as a baseline to understand the expected degradation in model capabilities.

5) Although I do think the work shows promise and is in a publishable state, addressing these weaknesses would add to the claims presented.



[1] Zhang, Fred, and Neel Nanda. "Towards best practices of activation patching in language models: Metrics and methods." arXiv preprint arXiv:2309.16042 (2023).
[2 Hanna, Michael, Ollie Liu, and Alexandre Variengien. "How does GPT-2 compute greater-than?: Interpreting mathematical abilities in a pre-trained language model." Advances in Neural Information Processing Systems 36 (2023): 76033-76060.]

**Questions:**

Have you performed any experiments on the attention weights of the models?
In general, do we find these neurons in the early/middle/later layers of the models?

---

> ### Author Response · Authors · 2025-11-19
>
> We sincerely appreciate your positive evaluation and constructive feedback on our work. You highlighted the clarity of writing, comprehensiveness of experiments, and robustness of the proposed methodology, as well as its broad validation across models and architectures. We have carefully reviewed and fully understood your suggestions for improvement, and we provide detailed responses and additional clarifications below. In particular, to address your main concerns regarding the novelty and the depth of layer-wise analysis, we have added new comparative experiments and extended analyses in the **Comparison Experiments section and Appendix A.9 (blue font)** to strengthen the completeness and interpretability of our arguments. Below, we respond in detail to each of the concerns and questions you raised.

---

> ### Author Response · Authors · 2025-11-19
> **Part 1/3**
>
> ## Weakness 1
>
> Most of reviewers have noted the need to clarify the distinction between our work and *Yu et al.* (2025) [1].
>
> We have clarified this distinction in the revised ***Related Work* (red font)** section. While both studies examine how a very small subset of parameters or units can disproportionately affect overall model performance, they differ fundamentally in **research objectives, disruption magnitude, methodological design, mechanism discovery, and the content and process of the conclusions**:
>
> 1. Disruption strength and effect level: *Yu et al.* (2025) analyzes individual **weights**, where perturbations increase perplexity by up to 3 orders of magnitude and only reduce task accuracy to random guessing. Our work targets **neuron-level lesions**, where masking just 3–7 neurons increases perplexity by up to **20 orders of magnitude** and entirely destroys task understanding. This reveals a deeper **functional collapse** rather than a numerical perturbation.
> 2. Methodological scalability and engineering applicability: The method of *Yu et al.* (2025) relies on manual observation to identify super weights, limiting reproducibility. Our proposed **perturbation–causal verification framework** automatically and systematically locates critical neurons, forming a standardized, reproducible pipeline that can be directly applied in engineering practice to identify and protect these critical neurons.
> 3. Discovery of multi-neuron interaction mechanisms: We find that collapse is not triggered by a single “super neuron,” but by the **collective interaction of a small group of critical neurons**, Any subset alone fails to cause comparable damage (see Ablation Study purple font), exhibiting a **phase-transition** phenomenon. *Yu et al.* (2025) does not explore such multi-unit interaction or nonlinear catastrophic mechanisms.
> 4. Different research units: *Yu et al.* (2025) investigates **parameter-level** importance, reflecting primarily numerical amplification effects. In contrast, our analysis operates at the **neuron level**, uncovering the causal and functional roles of neurons and identifying structural bottlenecks underlying high-level language abilities.
> 5. Differences in conclusions and discovery process: Rather than focusing on numerical outliers, our findings emerge from analyzing functional collapse patterns, which reveal sharp phase transitions when coordinated neuron groups are disrupted.
>
>
>
> [1]Mengxia Yu, De Wang, Qi Shan, Colorado J Reed, Alvin Wan   **The Super Weight in Large Language Models**     https://arxiv.org/abs/2411.07191
>
>
>
> ## Weakness 2
>
> Thank you for this helpful comment and for pointing out the connection to prior perturbation-based work in LLM. We have revised the introduction accordingly to reflect these changes.
>
> 1. We have de-emphasized the neuroscience / lesion-study framing and now ground the method primarily in prior work.
>    We agree that an explicit analogy to lesion experiments in neuroscience is not necessary to motivate our method. In the revised manuscript, we removed the lesion-style wording, and neuroscience now appears only in the introduction as brief historical context for the question of whether sparse, causally important “critical neurons” exist.
>
> 2. We now explicitly motivate our approach using perturbation-based methods from the AI literature[1, 2].
>    Following the suggestion, we updated the introduction so that the methodological framing is based on existing perturbation-based analyses of LLM.
>
> The revised manuscript follows the reviewer’s recommendation: the neuroscience / lesion analogy has been minimized and is no longer needed to justify the method. The main conceptual and methodological framing now comes from perturbation-based work on LLMs..
>
> [1] Zhang, Fred, and Neel Nanda. "Towards best practices of activation patching in language models: Metrics and methods." arXiv preprint arXiv:2309.16042 (2023).
>
> [2] Hanna, Michael, Ollie Liu, and Alexandre Variengien. "How does GPT-2 compute greater-than?: Interpreting mathematical abilities in a pre-trained language model." NIPS24

---

> ### Author Response · Authors · 2025-11-19
> **Part 2/3**
>
> ## Weakness 3
>
> To address the concern you raised, we conducted additional experiments on several models and reached the following conclusion: **neurons in the early MLP layer (UP_PROJ) and the middle MLP layer (GATE_PROJ) do not contribute significantly to model degradation.** We have added this analysis to the revised paper on ablation study (blue font).
>
> As shown in **Table 1** and **Table 2**, we performed random-masking experiments on the UP_PROJ and GATE_PROJ layers within the MLP blocks that contain the critical neurons. Each experiment was repeated 100 times with masking sizes of {10, 100, 1000}. The tables report the perplexity (mean ± standard deviation) on 1,000 WikiText-103 samples after masking. We observe that **randomly masking neurons in UP_PROJ and GATE_PROJ does not cause model collapse**, but only leads to small numerical fluctuations.
>
> Furthermore, as shown in **Table 3**, to avoid the possibility that random masking fails to select the truly important neurons in these layers, we compute importance scores for all neurons in UP_PROJ and GATE_PROJ following Algorithm 1. We then mask the top-10, top-100, and top-1000 neurons ranked by importance. We find that **even targeted masking of the most important neurons in these layers does not cause catastrophic failure**, and again only results in numerical degradation.
>
> #### Table 1. UP_PROJ random masking experiment
>
> | Model                 | baseline | 10             | 100            | 1000              |
> | --------------------- | -------- | -------------- | -------------- | ----------------- |
> | Qwen2.5-0.5B-Instruct | 18.1606  | 18.1832±0.0155 | 18.3729±0.1053 | 20.5223±0.3488    |
> | Llama-3.2-1B-Instruct | 17.9204  | 17.9223±0.0077 | 18.3061±1.1532 | 112.8353±191.3604 |
> | Llama-3.2-3B-Instruct | 13.4696  | 13.4713±0.0036 | 13.7866±0.9723 | 24.6764±18.9777   |
>
> #### Table 2. GATE_PROJ random masking experiment
>
> | Model                 | baseline | 10      | 100     | 1000     |
> | --------------------- | -------- | ------- | ------- | -------- |
> | Qwen2.5-0.5B-Instruct | 18.1606  | 40.2923 | 40.1173 | 43.1215  |
> | Llama-3.2-1B-Instruct | 17.9204  | 73.5823 | 77.6493 | 307.3553 |
> | Llama-3.2-3B-Instruct | 13.4696  | 16.5175 | 16.5206 | 497.5569 |
>
> #### Table 3. Masking UP_PROJ and GATE_PROJ neurons based on Imp(s)
>
> | Model                 | baseline | 10      | 100     | 1000     |
> | --------------------- | -------- | ------- | ------- | -------- |
> | Qwen2.5-0.5B-Instruct | 18.1606  | 40.2923 | 40.1173 | 43.1215  |
> | Llama-3.2-1B-Instruct | 17.9204  | 73.5823 | 77.6493 | 307.3553 |
> | Llama-3.2-3B-Instruct | 13.4696  | 16.5175 | 16.5206 | 497.5569 |
>
>
>
> ## Weakness 4
>
> In the main paper, we already included a random-masking baseline under the same masking size as the critical-neuron cluster (see paper's Figure 3), which shows that masking an equal number of randomly selected neurons does not harm the model. Since you raised this point again, we provide a more detailed and explicit presentation here.
>
> To further clarify this comparison, we conduct additional experiments using random neuron clusters of the same size as the identified critical-neuron set. As shown in **Table 4**, we run 100 random trials on 1,000 WikiText samples and report the corresponding perplexity values (mean ± standard deviation).
>
> #### Table 4
>
> | Model                 | neuron number | baseline | random           |
> | --------------------- | ------------- | -------- | ---------------- |
> | Qwen2.5-0.5B-Instruct | 3             | 18.1606  | 18.1671 ± 0.0331 |
> | Llama-3.2-1B-Instruct | 4             | 17.9204  | 17.9232 ± 0.0056 |
> | Llama-3.2-3B-Instruct | 4             | 13.4696  | 13.4713 ± 0.0071 |

---

> ### Author Response · Authors · 2025-11-19
> **Part 3/3**
>
> ## Question
>
> For this question. We provide detailed experimental evidence to address both aspects of your concerns.
>
> First, regarding whether neurons in the Attention layers could also trigger catastrophic degradation, we conducted a series of experiments on the Attention blocks corresponding to the same layers in which the critical neurons were identified. As shown in **Table 5**, we randomly masked neurons in these Attention layers with masking sizes of 10, 100, and 1000. The table reports perplexities (mean ± standard deviation) over 1,000 WikiText samples. The results show that masking Attention neurons does **not** lead to model collapse; the changes remain small and primarily numeric.
>
> Similarly, to avoid the possibility that random masking may miss the most influential Attention neurons, we also applied our importance-score–based selection to the Attention layers. As shown in **Table 6**, masking the top 10, top 100, and top 1000 Attention neurons ranked by importance scores likewise results in only mild degradation, confirming that Attention neurons are **not** responsible for the catastrophic failure.
>
> #### Table 5: Random masking in Attention layers
>
> | Model                 | baseline | 10             | 100            | 1000            |
> | --------------------- | -------- | -------------- | -------------- | --------------- |
> | Qwen2.5-0.5B-Instruct | 18.1606  | 18.1861±0.0206 | 18.4337±0.1260 | 28.3166±14.7690 |
> | Llama-3.2-1B-Instruct | 17.9204  | 18.0143±0.1495 | 18.5370±0.5181 | 28.1839±10.2511 |
> | Llama-3.2-3B-Instruct | 13.4696  | 13.4716±0.0130 | 13.5431±0.0694 | 14.0995±0.2346  |
>
> #### Table 6: Importance-score-based masking in Attention layers
>
> | Model                 | baseline | 10      | 100     | 1000     |
> | --------------------- | -------- | ------- | ------- | -------- |
> | Qwen2.5-0.5B-Instruct | 18.1606  | 18.1616 | 24.0459 | 109.2679 |
> | Llama-3.2-1B-Instruct | 17.9204  | 17.9034 | 19.1652 | 24.3669  |
> | Llama-3.2-3B-Instruct | 13.4696  | 13.4673 | 13.8156 | 14.8251  |
>
>
>
> Second, regarding the distribution of critical neurons across layers, we clarify that in the main paper we state that critical neurons primarily appear in the “outer layers” (see Section 4.2 "***Architectural Concentration: Critical Neurons Cluster in Outer Layers and MLP Compo-***
> ***nents***" and paper's Figure 2). We clearly define the layer groups: the first 33% of layers are **early layers**, the middle 34% are **middle layers**, and the last 33% are **late layers**.
>
> The distribution of critical neurons across these three regions is shown in **Table 7**. Across 21 models, we observe that **early layers contain on average 73.6%** of all critical neurons, **late layers contain 25.7%**, while **middle layers contain only 0.7%** (and only Gemma-2B shows middle-layer critical neurons at all).
>
> #### Table 7: Distribution of critical neurons across early / middle / late layers
>
> | Model                         | Early (%) | Middle (%) | Late (%) |
> | ----------------------------- | --------- | ---------- | -------- |
> | Llama-3.2-1B-Instruct         | 100       | 0          | 0        |
> | Llama-3.2-3B-Instruct         | 100       | 0          | 0        |
> | Llama-3-8B-Instruct           | 60        | 0          | 40       |
> | Llama-3.3-70B-Instruct        | 42.9      | 0          | 57.1     |
> | Gemma-2B                      | 85.7      | 14.3       | 0        |
> | Gemma-7B                      | 66.7      | 0          | 33.3     |
> | Deepseek-R1-Distill-Qwen-1.5B | 52.2      | 0          | 47.8     |
> | …                             | …         | …          | …        |
> | **Average**                   | **73.6**  | **0.7**    | **25.7** |

---

> > ### Comment · Reviewer_AABL · 2025-11-24
> > **Response to Official Comments**
> >
> > Thank you for your detailed response to the review. The added experimentation and clarifications are great. I have increased my score to reflect how these changes affect the contribution of the paper. Although, as a comment, I would recommend the authors to put more focus on the novelty of the contribution as prior literature overlaps with your findings in the final version of the paper.

---

> > > ### Author Response · Authors · 2025-11-25
> > >
> > > Thank you for your helpful feedback and for increasing your score. We appreciate your suggestion. We will follow your suggestion to prepare the final version. Thanks again!

---

### Meta-Review · Area_Chair_rRmu · 2026-01-07

**Summary:**

The paper argues that LLMs contain ultra-sparse critical neuron sets that are causally necessary for core language ability, and it proposes a two-stage Perturbation-based Causal Identification pipeline (sensitivity ranking + greedy masking) to find them across many architectures and scales. The main strengths are the breadth of experiments across multiple models and benchmarks, a clear algorithmic description with ablations and baselines. There are some concerns regarding the novelty - considering the existing works on identifying important neurons and the limited mechanistic explanation beyond careful characterization. Overall, I lean towards accepting it.

**Reviewer Concerns:**

The rebuttal substantially addressed the main methodological and validity concerns from reviewers Hd7S, Jfe4, and AABL. The main remaining outstanding concern is from SYxg: he/she still does not find the phenomenon sufficiently significant without a deeper mechanism/positioning versus related work.

**Reviewer Scores:**

Based on explicit follow-up comments, the final scores for this paper would be 8, 8, 6, 2.

---

### Decision · Program_Chairs · 2026-01-26

Accept (Poster)